# Geographic variation in the advertisement calls of *Hyla eximia* and its possible explanations

Ruth E. Rodríguez-Tejeda[1], María Guadalupe Méndez-Cárdenas[2], Valentina Islas-Villanueva[3] and Constantino Macías Garcia[1]

[1] Instituto de Ecología, Universidad Nacional Autónoma de México, México, Mexico
[2] Instituto de Investigaciones Antropológicas, Universidad Nacional Autónoma de México, México, Mexico
[3] Instituto de Ciencias del Mar y Limnología, Universidad Nacional Autónoma de México, México, Mexico

## ABSTRACT

Populations of species occupying large geographic ranges are often phenotypically diverse as a consequence of variation in selective pressures and drift. This applies to attributes involved in mate choice, particularly when both geographic range and breeding biology overlap between related species. This condition may lead to interference of mating signals, which would in turn promote reproductive character displacement (RCD). We investigated whether variation in the advertisement call of the mountain treefrog (*Hyla eximia*) is linked to geographic distribution with respect to major Mexican river basins (Panuco, Lerma, Balsas and Magdalena), or to coexistence with its sister (the canyon treefrog, *Hyla arenicolor*) or another related species (the dwarf treefrog, *Tlalocohyla smithii*). We also evaluated whether call divergence across the main river basins could be linked to genetic structure. We found that the multidimensional acoustic space of calls from two basins where *H. eximia* currently interacts with *T. smithii*, was different from the acoustic space of calls from *H. eximia* elsewhere. Individuals from these two basins were also distinguishable from the rest by both the phylogeny inferred from mitochondrial sequences, and the genetic structure inferred from nuclear markers. The discordant divergence of *H. eximia* advertisement calls in the two separate basins where its geographic range overlaps that of *T. smithii* can be interpreted as the result of two independent events of RCD, presumably as a consequence of acoustic interference in the breeding choruses, although more data are required to evaluate this possibility.

## INTRODUCTION

Populations of species occupying large geographic ranges are likely to experience different selective pressures (*West-Eberhard, 1983*; *Panhuis et al., 2001*; *Coyne & Orr, 2004*) which, together with drift (*Wiens, 2004*), may result in phenotypic and genotypic differences between populations (e.g., *Avise, 2000*; *Laugen et al., 2003*; *Amézquita et al., 2009*). Local

Corresponding author
Constantino Macías Garcia,
maciasg@unam.mx

differences in ecology, such as prey availability (e.g., *Arnold, 1980*; *Bonansea & Vaira, 2007*) or habitat structure (e.g., *Relyea, 2002*; *Skelly, 2004*) can lead to differential adaptation between populations (*Newman, 1992*), but in species with generalized diets and habitat requirements such variation would have a limited effect on differentiation (e.g., *Virgós, Llorente & Cortés, 1999*). On the other hand, environmental variation in the attributes that determine the transmission and reception of signals used in social (*Wells, 1977*; *Sullivan & Wagner, 1988*; *Wagner, 1989*) or sexual contexts (*Searcy & Andersson, 1986*) may lead to rapid population differentiation (*Jennions & Petrie, 1997*; *Seehausen, van Alphen & Witte, 1997*; *Lougheed et al., 2006*; *Boul et al., 2007*). This is because signals used by animals in a breeding context may convey information about the species, sex, breeding status, and even the condition of the sender (*Gerhardt, 1992*; *Wilczynski & Chu, 2001*), all of which may be relevant to conspecifics searching for mating partners (*Butlin & Ritchie, 1994*; *Emerson, 2001*; *Forsman & Hagman, 2006*). Geographic variation in mating signals has been widely reported in studies of character displacement, where specific traits (e.g., morphological, behavioral, ecological or physiological traits) differ among sympatric and allopatric populations due to the risk of maladaptive hybridization with related species (*Brown & Wilson, 1956*; *Grant, 1972*). Reproductive character displacement (RCD) is often studied in systems where sister species meet at typically narrow hybrid or tension zones (*Butlin, 1987*; *Howard, 1993*; *Butlin & Ritchie, 1994*). However an approach involving the study of signal variation across wide geographic areas, coupled with genetic and geographic data, could be helpful in identifying evolutionary patterns of signal evolution and tracing the links between micro and macro evolution of mating signals (*Avise et al., 1987*).

Amphibians are good models for studying signal evolution and reproductive isolation since their dispersal is restricted by habitat, and they thus are highly philopatric, facilitating the study of genetic divergence amongst populations (c.f. *Gamble, MacGarigal & Compton, 2007*). Amphibian philopatry results in populations being genetically structured over short geographic distances, which facilitates inferences about the historic events that caused their current distribution (*Zeisset & Beebee, 2008*). Although other modalities (e.g., visual; *Hartmann et al., 2005*; *Reynolds & Fitzpatrick, 2007*; *Taylor et al., 2008*) are sometimes involved, it is common in anurans that species recognition and female preference depend on a single acoustic signal; the advertisement call *Wells, 1977*; *Cocroft & Ryan, 1995*; *Wells & Schwartz, 2007*. This is produced by males, typically during the mating season. Several attributes of the calls vary within a species as a function of male morphology and condition. For instance, the frequency or pitch (Hertz) of the call is inversely correlated with body size (*Zug, 1993*; *Gerhardt & Huber, 2002*). Calls are also affected by temperature, which influences the length of the call as well as the number of pulses that compose the call, and the pulse rate (*Gerhardt & Davis, 1988*; *Wilczynski & Chu, 2001*; *Gerhardt & Huber, 2002*; *Yamaguchi et al., 2008*). Additionally, males at a chorus often adjust the timing of their calls to avoid masking by other vocalizations, and when choruses are formed by more than one species, acoustic interference can select for the use of different acoustic space (*Brush & Narins, 1989*; *Schwartz, Buchanan & Gerhardt, 2002*; *Chek, Bogart & Lougheed, 2003*; *Hoskin & Higgie, 2010*). Acoustic space is a multidimensional space defined by

**Peer**J

acoustic properties of the signals, such as frequency, duration, amplitude, temporal structure, etc.; see, for instance, *Ryan & Rand (2003)*. In spite of variation, anuran songs are sufficiently stereotyped to permit females to identify and assess males (e.g., the *Hylidae*; *Gerhardt, 1991*; *Gerhardt, 1992*), so much so that *Hyla* spp. have often been used as models to study pre-mating reproductive isolation amongst sister species (e.g., *Littlejohn, 1965*; *Ball & Jameson, 1966*; *Littlejohn, 1970*; *Duellman, 1973*; *Gerhardt, 1994*; *Höbel & Gerhardt, 2003*; *Gerhardt, 2005*).

*Hyla eximia* is a relatively small (ca. 3 cm) tree-frog endemic to the geologically complex Trans-Mexican Volcanic Belt (TMVB; *de Cserna, 1989*; *Ferrari et al., 2012*) and adjacent Mexican High Plateau (*Duellman, 1970*; *Duellman, 2001*). It gives its name to the *H. eximia* species group (*Smith et al., 2007*), which includes eleven other species (*Faivovich et al., 2005*). The very similar *H. wrightorum* replaces *H. eximia* to the north (through Arizona), and *H. euphorbiacea* is found to the south of the TMVB (allopatric with *H. eximia*). *H. plicata* is endemic to the TMVB and is partially sympatric with *H. eximia* (*Smith et al., 2007*), and *H. arenicolor* is sympatric with *H. eximia* in parts of the Mexican Plateau. Like *H. eximia*, *H. arenicolor* has a wide geographic range; in the Mexican Plateau it is mainly found in the mountain zones of Balsas Basin, at elevations of 300–3000 m (*Duellman, 2001*). *Tlalocohyla smithii* (formerly *Hyla smithii*, *Faivovich et al., 2005*) is distributed in the Pacific lowlands of Mexico up to elevations of about 1000 m, from central Sinaloa to southern Oaxaca, and inland within the Balsas Basin (*Duellman, 2001*). *Tlalocohyla smithii*, is not part of the *H. eximia* group, but occupies the same matings pools.

Previous work has shown evidence of geographic variation in *H. eximia* calls. Indeed, variation in pulse rate, call duration and the dominant frequency among recordings from several populations (*Blair, 1960*; *Duellman, 1970*; *Sullivan, 1986*; *Duellman, 2001*) was so great that *Duellman (1970)* and *Duellman (2001)* concluded that the species lacks a typical call, though sample sizes per population were small. It has also been suggested that some variation in advertisement calls of *H. eximia* may be linked to RCD. Based on phonetic data *Cortés-Soto (2003)* suggested that *H. eximia* and *H. plicata* have evolved different advertisement calls in the 500 m altitudinal band (2400–2900 m asl) where their ranges overlap. There, the calls of *H. eximia* are shorter and contain fewer pulses than when the species are found in allopatry, suggesting that RCD has occurred. Here we describe the variation of the advertisement calls of *H. eximia* across a substantial part of its geographic range, and explore whether this variation is linked to genetic structure (based on mitochondrial and nuclear DNA sequences), geography (i.e., hydrographic basins) and/or range overlaps with its sister species, the canyon treefrog (*H. arenicolor*), and a related species, the Mexican dwarf treefrog (*Tlalocohyla smithii*).

## MATERIALS AND METHODS

### Song description

Based on information from public databases we surveyed 51 locations where *H. eximia* has been previously collected. We found *H. eximia* in nine of these locations. Surveys were carried out during a single rainy season to avoid bias due to inter-seasonal call variation

 

**Peer**J

**Table 1  Populations sampled.** Populations where advertisement calls of *Hyla eximia*, *Hyla arenicolor* and *Tlalocohyla smithii* were recorded.

| Population | Symbol map | Latitude | Longitude | Chorus composition | N |
|---|---|---|---|---|---|
| El Realejo, San Luis Potosí | R | 22°40′23.98″N | −100°25′5.45″W | *H. eximia* allopatric | 26 |
| Laguna de Gerardo, San Luis Potosí | L | 22°38′27.36″N | −100°26′20.94″W | *H. eximia* allopatric | 29 |
| Sierra de Álvarez, San Luis Potosí | SA | 22°01′52.56″N | −100°36′49.86″W | *H. eximia* allopatric | 15 |
| Playita de San Rafael, Jalisco | SR | 20°02′52.98″N | −103°09′46.26″W | *H. eximia* allopatric | 25 |
| Rancho Santa Elena, Hidalgo | SE | 20°08′7.38″N | −98°30′43.50″W | *H. eximia* sympatric with *H. arenicolor* | 18 |
| Magdalena, Jalisco | MA | 20°54′0.41″N | −103°59′36.85″W | *H. eximia* sympatric with *T. smithii* | 20 |
| Presa de Malinaltenango, Edo. de México | PM | 18°48′32.01″N | −99°43′26.18″W | *H. eximia* sympatric with *T. smithii* | 7 |
| Rancho Santa Elena, Hidalgo | SE | 20°08′7.38″N | −98°30′43.50″W | *Hyla arenicolor* | 2 |
| Km 58, San Luis Potosí | K58 | 22°01′12.18″N | −100°36′7.26″W | *Hyla arenicolor* | 7 |
| Magdalena, Jalisco | MA | 20°54′0.41″N | −103°59′36.85″W | *Tlalocohyla smithii* | 12 |
| Presa de Malinaltenango, Edo. de México | PM | 18°48′32.01″N | −99°43′26.18″W | *Tlalocohyla smithii* | 5 |
| Tecomatepec, Edo. de México | TC | 18°50′9.06″N | −99°41′40.62″W | *Tlalocohyla smithii* | 3 |

and to ensure the presence of the species in the breeding pools. *Hyla eximia* was alone (allopatric) in four of the nine locations that were sampled. In two locations, *H. eximia* was sympatric and formed breeding choruses with *H. arenicolor*, and in the remaining three with *T. smithii* (Table 1, Fig. 1). Here we considered "sympatric" the populations in which two species were found to simultaneously occupy the same microhabitat during the breeding season (form breeding choruses); this preliminary classification that will have to be confirmed with repeated sampling.

During the summer of 2011 we recorded advertisement calls from males of the three species. Calls were recorded in natural breeding aggregations with a directional microphone (Sennheiser[TM]ME66) connected to a digital recorder (Marantz[TM]PMD660). The microphone was held 1 m in front of the calling male, and the recording volume was adjusted to avoid saturation (0–6 dB). Each male was recorded for 1–1.5 min to ensure that adequate call series were obtained. Then we measured the body temperature of the frog by holding it from a hind limb and pointing an infrared thermometer (Extech[TM]42529; ±0.05 °C) at its body, thus preventing heat transfer from the observer. Finally, we measured the frog's snout-vent length with a digital calliper (±0.05 mm) and collected a toe-clip from the distal phalanx of the fourth right front-leg digit for subsequent DNA extraction (see *Gonser & Collura, 1996*).

We used only recordings containing at least 10 advertisement calls. In order to maximize the opportunity to detect differences in the songs of *H. eximia* between populations, we measured a large number of variables from our recordings. Using Avisoft-SASLab Pro[TM] we quantified the following attributes (definition, abbreviation; units): call duration (length of the call, continuous trace in the sonogram, CD; s), inter-call duration (silent

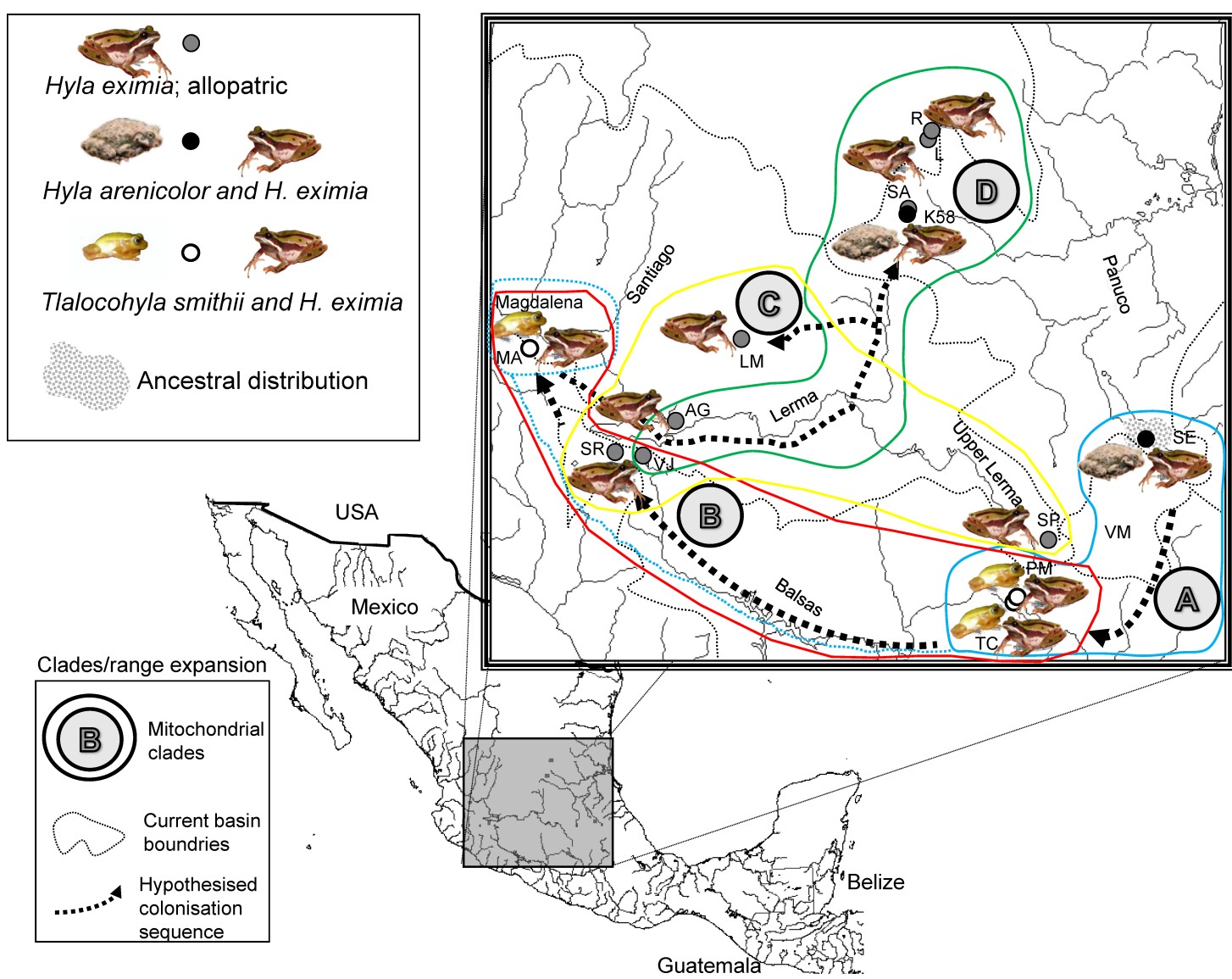

**Figure 1 Localities sampled, chorus composition and mitochondrial clades.** Hypothesized colonization routes of *H. eximia* in central Mexico. The distribution of frogs belonging to clades A–D from a phylogenetic analysis of two concatenated mitochondrial genes (see text) is indicated with globes of different color (clade A, turquoise blue; clade B, red; clade C, yellow; D, green). Dotted arrows indicate the possible direction of the colonization, and are not intended to show the exact paths followed by the frogs. The occurrence of one individual from Magdalena in clade A is indicated with a dotted turquoise-blue line. Thin grey dotted lines represent boundaries between the major basins (Panuco, Lerma, Balsas and the currently closed Valley of Mexico), whereas rivers are indicated with continuous grey lines. Lerma River flows northwards until it meets the Santiago, which drains in the Pacific Ocean to the west. Balsas drains southwards, also in the Pacific, whereas the Panuco drains to the east, in the Gulf of Mexico. See Table 1 for population codes.

period between consecutive calls, IC; s), call period (sum of CD + IC, CP; s), number of pulses (number of short signals that compose the call, NP; count), pulse duration (length from the beginning to the end of the pulse, PD; s), inter-pulse duration (length of the silent period between the end of one pulse and the beginning of the next, ID; s), pulse amplitude (peak intensity of the pulse, PA; V), pulse peak frequency (highest energy within the

pulse, PPF; kHz), pulse period (the sum of PD + IP, PP; s), pulse rate (pulses per second measured from the start of the first pulse to the start of the last pulse, PR; Hz), dominant frequency (frequency with the highest energy within the call, DF; kHz) and fundamental frequency (the lowest, or reference frequency of series of frequencies -called harmonics- that are its integer multiples, FF; kHz). Temporal variables were measured directly on the spectrogram, all the pulse variables were extracted with the *Pulse Train Analysis* option, and the frequency variables were obtained with *Power spectrum logarithmic* function (Fig. S1). Individual frogs were represented in the analyses by the average value of each call attribute.

To detect the call variation in *H. eximia* we defined three chorus compositions: (1) allopatric, composed of *H. eximia* males only, (2) *H. eximia* in mixed choruses with *H. arenicolor*, and (3) *H. eximia* in mixed choruses with *T. smithii*. We also analyzed the calls of (4) *H. arenicolor* males in mixed choruses with *H. eximia*, and (5) *T. smithii* males in mixed choruses with *H. eximia*.

## Call variation

To reduce the number of variables included in multivariate analyses in an objective way, we calculated for each chorus composition the correlation matrix of the twelve variables and discarded one attribute of any pair that had a correlation coefficient of $\geq 0.7$. Since temperature is known to affect several amphibian call attributes (*Blair, 1958*; *Zweifel, 1959*; *Zweifel, 1968*; *Gerhardt & Mudry, 1980*; *Gayou, 1984*; *Gerhardt & Huber, 2002*) we performed an analysis of covariance (ANCOVA) for each species and variable to assess the potential effect of temperature on call attributes from different chorus compositions. Since ANCOVAs showed a significant effect of temperature on call variables, we used ANCOVA residuals for all subsequent analyses.

Once the number of variables had been reduced and the effect of temperature removed, we compared the attributes of the calls of *H. eximia* from the three chorus compositions using a multivariate analysis of variance (MANOVA) to determine whether and to what extent call variation was determined by locality and/or chorus composition. The absence of a significant effect of locality on call attributes might suggest that song variation occurs at a larger geographical scale, such as by basin, thus when there was no locality effect we repeated the analysis replacing locality with major river basins (Panuco, Lerma, Magdalena and Balsas). All analyses were performed using R (*R Development Core Team, 2011*). The above analyses were designed to evaluate whether the vocalizations of *H. eximia* produced at different chorus compositions were different in some -or in a combination of- attributes. To visualize such differences we performed a canonical discriminant analysis on the calls of *H. eximia,* and used the resulting discriminant function to plot the distribution of the calls of *H. arenicolor* and *T. smithii* in the same canonical space. This allowed the visualization of any significant differences revealed by the MANOVA as displacements away the calls of sympatric species in a canonical multivariate space. Finally we ran an analysis of variance to compare the first two canonical scores of *H. eximia* amongst basins—a major geographic source of variation. Analyses in this section were conducted with NCSS[TM] (Kaysville, UT).

## Genetic variation

Using mitochondrial and nuclear genes we analyzed phylogenetic patterns to explore the extent to which geographic variation of songs across populations is related to genetic variation. Total genomic DNA of 30 individuals of *Hyla eximia* from 12 localities was extracted from EtOH-preserved toe clippings using the DNeasy Tissue Kit (Qiagen™Inc., Valencia, CA, USA). Standard PCRs were carried out in 25 μl reaction mixes with final concentrations of 1 μM of each primer, 1.5 mM of $MgCl_2$, 0.2 mM of each dNTP, 1xNH4 reaction buffer (50 mM Tris–HCl pH 8.8; 16 mM $[NH4]_2 SO4$), 1.25 units of Taq DNA polymerase, and 1–4 μl of DNA. Successful amplifications were performed using the following protocol: initial denaturing for 5 min at 95 °C, denaturing for 45 s at 94 °C, annealing for 45 s at 48 °C (for Cyt b) and 57 °C (for ATP), extension for 1 min at 72 °C, final extension min for 7 min at 72 °C; denaturing, annealing and the first extension stage were cycled 35–40 times.

We amplified two mitochondrial gene regions; cytochrome-*b* (754 bp) with the primers MVZ16-H AAATAGGAARTATCAYTCTGGTTTRAT and MVZ15-L GAAC-TAATGGCCCACACWWTACGNAA (*Moritz et al., 1992*; *Stöck et al., 2008*; *Stöck et al., 2011*), and ATPase subunits 8 and 6 (778 bp) with the primers LysAF CAACCAC-CCTTGATGAATGCC and C3FR GGGCTGGGGGTTKACTATGTG (*Stöck et al., 2008*; *Bryson et al., 2010*; Table S2). These mitochondrial gene regions have been found to be informative at different levels of variation within hylid frogs (references given above). We also amplified two nuclear gene regions using the same protocol as above, with a slight modification (see below). The Pro-opiomelanocortin gene region (POMC; 447 pb) with the primers POMC_DRV_F1 ATATGTCATGASCCAYTTYCGCTGGAA and POMC_DRV_R1 GGCRTTYTTGAAWAGAGTCATTAGWGG (annealing temperature = 58 °C; modified from *Wiens et al., 2005*; *Bryson et al., 2010*), and the Rhodopsin (Rho) gen region (291 pb) with the primers Rho1A ACCATGAACGGAACAGAAGGYCC and Rho1C CCAAGGGTAGCG AAGAARCCTTC (*Bossuyt & Milinkovitch, 2000*; *Faivovich et al., 2005*; *Klymus et al., 2010*; Table S2).

PCR products were sent for sequencing to the University of Washington High-Throughput Genomics Unit, Department of Genome Sciences, to the Service of DNA Sanger Sequencing. Sequences were aligned with Clustal X (*Jeanmougin et al., 1998*). Some regions were difficult to align, and were therefore adjusted manually using BioEdit (*Hall, 1999*). DNAsp (*Librado & Rozas, 2009*) was used to determine the number of unique haplotypes and invariable and polymorphic sites.

Phylogenetic analyses were performed for each gene using the maximum likelihood algorithm (ML; *Felsenstein, 1981*). We used the Hasegawa–Kishino–Yano model (HKY) − G + I (gamma distribution with invariable sites) for mitochondrial genes and the GTR + I + G for nuclear genes. In both analyses the nearest neighbor interchange (NNI) was used. The best model was identified with the program Modeltest 3.7 and MrModeltest modified version of Modeltest version 3.6 (*Posada, 2008*) using the Akaike information criterion (AIC).

We also constructed a haplotype network using TCS 2.1.1 for preliminary exploration of the phylogeographic relationships of the obtained haplotypes (*Clement, Posada & Crandall, 2000*). Genetic diversity was calculated as nucleotide ($\pi$) and haplotype (*h*) diversity for each of the sub-regions of the basins (northern, central and southern) using Arlequin 3.5.1.2 (*Excoffier & Lischer, 2010*), for both pairs of concatenated genes (mitochondrial and nuclear). We also calculated the population differentiation index $F_{ST}$ and its significance levels, between pairs of basins, using Arlequin 3.5.1.2 (*Excoffier & Lischer, 2010*). To visualize the differences in nuclear genes between basins, the genetic structure was investigated using Structure 2.3 (*Pritchard, Stephens & Donnelly, 2000*) with the Admixture model and correlated frequencies. A 500,000 burnin was employed and 5,000,000 generations of K 1 to 10 were performed with 10 iterations for each gene. We used the Evanno method to determine the true value of K (*Evanno, Regnaut & Goudet, 2005*) with the program Structure Harvester (*Earl & vonHoldt, 2012*). This analysis was employed only for nuclear genes, as it is not designed to find structure in linked or haploid genes.

### Phenotypic, genetic and geographical distance

We performed Mantel tests and partial Mantel tests with 1000 resamplings using R v. 2.15.1 (*R Development Core Team, 2011*) to evaluate the correlation between acoustic (phenotypic), genetic and geographic distances. To calculate the acoustic matrix we employed the distances between canonical variables for each basin obtained from a discriminant analysis. The geographic distance was constructed with the geo- reference points taken at each locality; as the basin could include several localities, an average geo-reference was employed. The genetic distance matrix was built using the $D_{xy}$ values.

The project was reviewed and approved by the Consejo Técnico de la Investigación Científica (CTIC) of UNAM.

## RESULTS

### Song description

We obtained 169 good quality recordings of male advertisement calls. These came from nine *Hyla arenicolor*, 20 *Tlalocohyla smithii* and 140 *Hyla eximia* from both sympatric and allopatric localities. A detailed description of the calls of *H. eximia* based on our complete sample is available in the Supplemental Information (see song description S1) along with descriptions of those of *H. arenicolor* and *T. smithii* which are also summarized in Tables 2 and 3. Briefly, the calls of *H. eximia* consist of a single note that lasts less than a quarter of a second, has a dominant frequency of about 2.6 kHz, and is composed of ca. 18 pulses. The call of *Hyla arenicolor* (Table 3) is roughly three times longer, (0.88 s), although it is composed of only a few more (about 20 pulses) but these pulses are longer and louder than the calls of *H. eximia*. Calls of *H. arenicolor* are lower-pitched than those of *H. eximia* (dominant frequency = 1.26 kHz). The structure of the call of male *T. smithii* is very different from those of *H. eximia* and *H. arenicolor*. Often the males emit calls with two elements; a long note followed by an extremely short one. Most individuals, however, produce more long than short notes, and in several recordings short notes are absent.

**Table 2 Descriptive data for *H. eximia*.** Descriptive data of morphology, body temperature and attributes of the advertisement calls of *H. eximia*.

| Variable (units) | *H. eximia* allopatric (*n* = 95) | | | *H. eximia* (sympatric with *H. arenicolor*) (*n* = 18) | | | *H. eximia* (sympatric with *T. smithii*) (*n* = 27) | | |
|---|---|---|---|---|---|---|---|---|---|
| | Mean | SD | CV | Mean | SD | CV | Mean | SD | CV |
| SLV (mm) | 28.149 | 2.426 | 8.618 | 32.252 | 1.734 | 5.376 | 31.011 | 3.197 | 10.311 |
| T (°C) | 17.634 | 2.041 | 11.572 | 14.478 | 1.663 | 11.489 | 19.556 | 3.226 | 16.497 |
| CD (s) | 0.199 | 0.015 | 7.417 | 0.267 | 0.019 | 7.018 | 0.196 | 0.025 | 12.907 |
| IC (s) | 0.401 | 0.124 | 30.826 | 0.548 | 0.133 | 24.344 | 0.657 | 0.416 | 63.282 |
| CP (s) | 0.599 | 0.123 | 20.58 | 0.815 | 0.127 | 15.563 | 0.853 | 0.432 | 50.669 |
| NP | 20.435 | 1.791 | 8.766 | 16.133 | 1.605 | 9.947 | 12.066 | 2.925 | 24.244 |
| PD (s) | 0.005 | 0 | 9.148 | 0.005 | 0 | 4.587 | 0.007 | 0.002 | 24.963 |
| ID (s) | 0.009 | 0.001 | 12.086 | 0.017 | 0.002 | 9.455 | 0.016 | 0.005 | 29.066 |
| PA (V) | 0.184 | 0.135 | 73.674 | 0.437 | 0.149 | 34.216 | 0.261 | 0.111 | 42.48 |
| PPF (kHz) | 2.380 | 2.147 | 9.023 | 2.260 | 0.079 | 3.5 | 2.760 | 0.484 | 17.51 |
| PP (s) | 0.014 | 0.001 | 9.381 | 0.021 | 0.002 | 7.367 | 0.023 | 0.006 | 24.969 |
| PR (Hz) | 98.139 | 9.776 | 9.961 | 56.641 | 5.196 | 9.174 | 57.241 | 15.683 | 27.398 |
| DF (kHz) | 2.590 | 0.207 | 8.004 | 2.410 | 0.135 | 5.61 | 3.003 | 0.618 | 20.59 |
| FF (kHz) | 0.649 | 0.293 | 45.19 | 0.647 | 0.216 | 33.4 | 0.534 | 0.222 | 41.5 |

**Notes.**

SVL, snout vent length; T, corporal temperature; CD, call duration; IC, inter-call duration; CP, call period; NP, number of pulses; PD, pulse duration; ID, inter-pulse duration; PA, pulse amplitude; PPF, pulse peak frequency; PP, pulse period; PR, pulse rate; DF, dominant frequency; FF, fundamental frequency.

Some males produced a third note with different structure and duration than the previous two, but it is not clear whether this is also part of the advertisement call. For our analyses we considered just the first note, as it is the most constant element in our recordings, and it is the one more likely to interfere with the advertisement calls of *H. eximia* calls. Spectrograms of typical advertisement calls of the three species are shown in Fig. 2.

## Call variation

The following variables were highly correlated in the calls of *Hyla eximia* in allopatry: pulse amplitude and interval between calls (0.997), and call period and inter-call duration (0.993). In sympatry with *H. arenicolor*, the correlated variables of calls of *H. eximia* were: call period and inter-call duration (0.991), pulse rate and inter-pulse duration (−0.984), pulse period and inter-pulse duration (0.954), and pulse rate and pulse period (−0.953). Correlated variables in the contact zone between *H. eximia* and *T. smithii* were: call period and inter-call duration (0.999), pulse period and pulse duration (0.996), pulse rate and inter-pulse duration (−0.966), dominant frequency and pulse peak frequency (0.943), and pulse rate and number of pulses (0.920).

The following attributes of the calls of *H. arenicolor* were intercorrelated: pulse rate and inter-pulse duration (−0.985), pulse period and inter-pulse duration (0.965), pulse rate and pulse period (−0.941), and call period and inter-call duration (0.917). In the case of *T. smithii* the highest correlations amongst variables were: call period and inter-call duration (0.998), pulse period and inter-pulse duration (0.990), pulse rate and inter-pulse duration (−0.979) and pulse rate and pulse period (−0.967).

**Table 3 Descriptive data for *H. arenicolor* and *T. smithii*.** Descriptive data of morphology, body temperature and attributes of the advertisement calls of *H. arenicolor* and *T. smithii*.

| Variable (units) | *H. arenicolor* (n = 9) | | | *T. smithii* (n = 20) | | |
|---|---|---|---|---|---|---|
| | Mean | SD | CV | Mean | SD | CV |
| SLV (mm) | 42.664 | 2.999 | 7.029 | 24.523 | 1.299 | 5.297 |
| T (°C) | 19.044 | 2.783 | 14.613 | 18.285 | 1.446 | 7.91 |
| CD (s) | 0.884 | 0.182 | 20.623 | 0.469 | 0.055 | 11.714 |
| IC (s) | 1.6 | 0.473 | 29.583 | 1.18 | 0.923 | 78.266 |
| CP (s) | 2.499 | 0.527 | 21.089 | 1.649 | 0.902 | 54.714 |
| NP | 22.613 | 4.453 | 19.693 | 23.735 | 4.935 | 20.794 |
| PD (s) | 0.008 | 0.003 | 42.701 | 0.005 | 0.001 | 9.823 |
| ID (s) | 0.039 | 0.007 | 16.904 | 0.02 | 0.004 | 16.95 |
| PA (V) | 0.368 | 0.209 | 56.86 | 0.214 | 0.152 | 71.076 |
| PPF (kHz) | 1.470 | 0.66 | 45.031 | 4.320 | 0.163 | 3.8 |
| PP (s) | 0.047 | 0.009 | 19.567 | 0.025 | 0.003 | 13.41 |
| PR (Hz) | 24.837 | 4.91 | 19.767 | 48.218 | 7.661 | 15.888 |
| DF (kHz) | 1.264 | 0.666 | 52.662 | 4.450 | 0.158 | 3.6 |
| FF (kHz) | – | – | – | 1.920 | 1.515 | 79.1 |

**Notes.**

SVL, snout vent length; T, corporal temperature; CD, call duration; IC, inter-call duration; CP, call period; NP, number of pulses; PD, pulse duration; ID, inter-pulse duration; PA, pulse amplitude; PPF, pulse peak frequency; PP, pulse period; PR, pulse rate; DF, dominant frequency; FF, fundamental frequency. The calls of *H. arenicolor* showed no fundamental frequency.

Considering that correlations ≥0.7 indicate redundant variables, we dropped from the analyses the following variables: call period (CP), pulse period (PP), pulse rate (PR), inter-pulse duration (ID) and pulse peak frequency (PPF). Consequently, analyses were performed using seven variables: call duration (CD), inter-call duration (IC), number of pulses (NP), pulse duration (PD), pulse amplitude (PA), dominant frequency (DF) and fundamental frequency (FF).

Covariance analyses revealed that temperature, chorus composition and the interaction between them affect call attributes of each species in different ways (Table S1). To control for the effect of temperature, in cases where the interaction between temperature and chorus composition was significant we used the residuals from the ANCOVA, rather than the original variable, in all subsequent analyses.

MANOVA results show that call variability in *H. eximia* is due to the locality ($F_{(6,42)} = 3.91, P = 1.66e^{-14}$), whereas the fact that individuals call in allopatry or sympatry does not contribute significantly to call variation. However, when populations are grouped according to watershed (Panuco, Lerma, Magdalena y Balsas), it is amongst basins, rather than localities, that calls diverge significantly ($F_{(3,21)} = 7.7241, P < 2e^{-16}$). The seven call variables entered in the MANOVA analyses allow for the discrimination between the four basins with a 34% reduction in classification error, classifying correctly 51% of the calls according to the basin of origin. The first two canonical functions explain 99.8% of the variance and define a multivariate acoustic space where the calls of

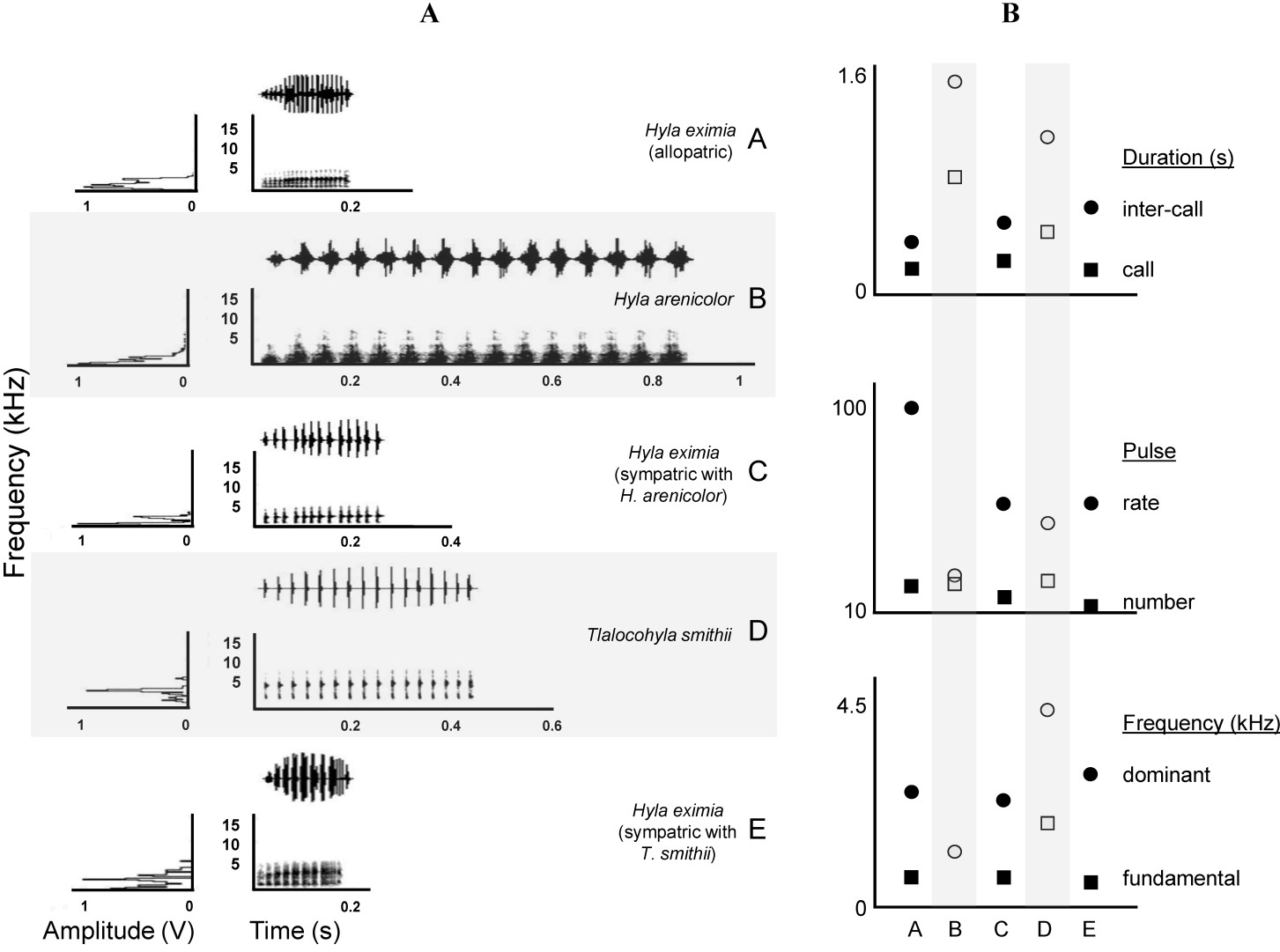

**Figure 2 Call spectrograms and averages of call attributes.** (A) Spectrograms of advertisement calls produced by individuals of each chorus composition (A, allopatric *Hyla eximia*; B, *Hyla arenicolor*; C, *H. eximia* sympatric with *H. arenicolor*; D, *Tlalocohyla smithii*; and E, *H. eximia* sympatric with *T. smithii*). (B) From top to bottom, mean temporal, pulse and frequency variables. Categories in the *X* axis correspond to the chorus composition. Variable identity, indicated next (right) to the solid symbols, apply also to the open symbols (e.g., circles of both types represent the same variable). Solid symbols indicate *H. eximia* populations and open symbols represents the heterospecific species.

*H. eximia* from Balsas and Magdalena are distinct from those from the other basins (Figs. 2 and 3). Subsequent ANOVAs and Tukey tests confirm that these differences are significant (function 1, $F_{3,134} = 87.67, P < 0.0001$; function 2, $F_{3,134} = 6.84, P = 0.0002$; Fig. 4). Discriminant function 1 distinguishes amongst calls from different basins based on the interval between calls ($r = -0.809$), call duration ($r = -0.274$) and number of pulses ($r = 0.254$), while the discriminant function 2 discriminates by dominant frequency ($r = 0.974$). Applying the functions that discriminate between calls of *H. eximia* from different basins to the call attributes (or the ANCOVA residuals) of *H. arenicolor* and

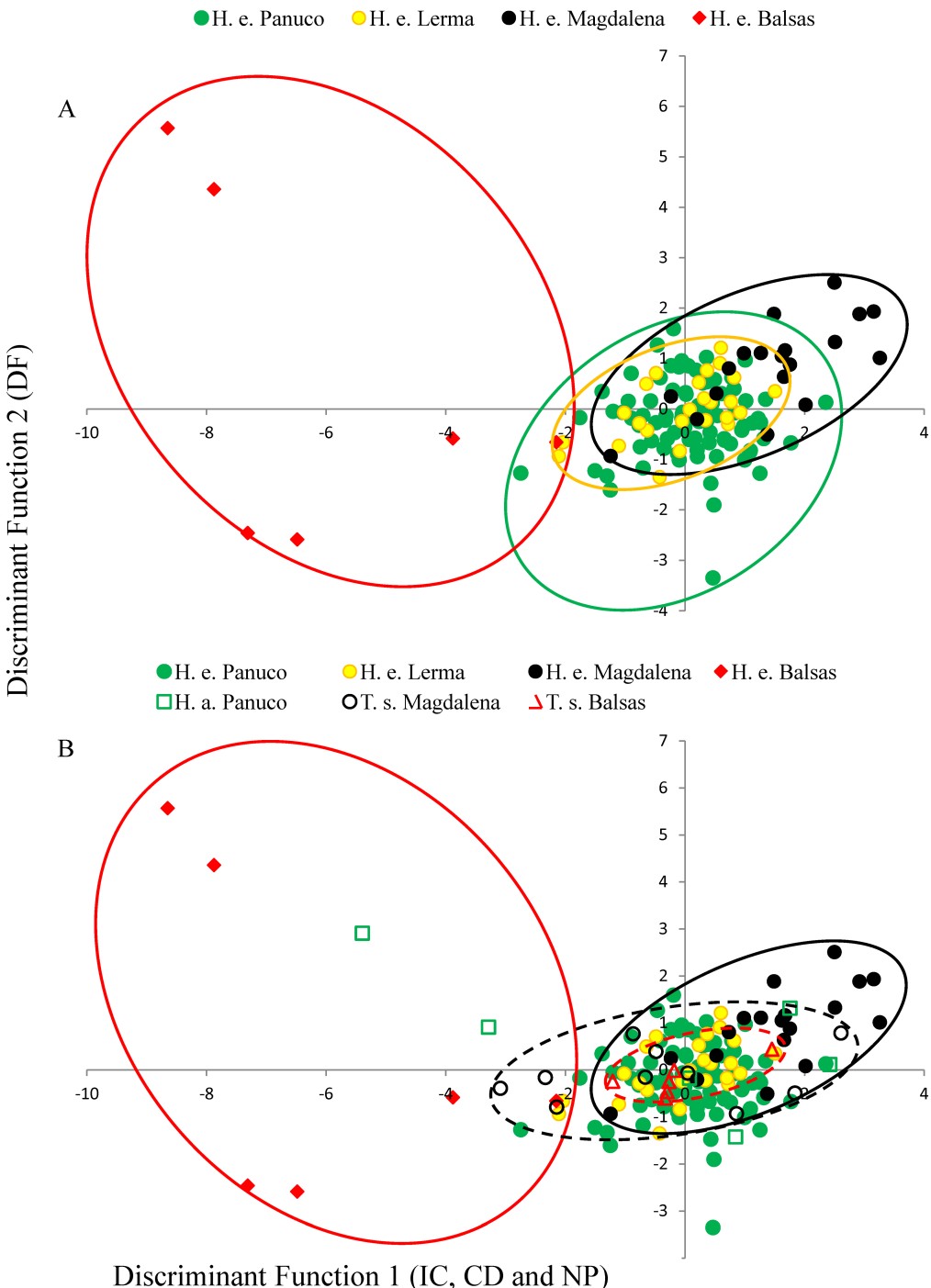

**Figure 3  Discriminant plot.** (A) Discriminant functions of advertisement call attributes of *Hyla eximia*. Discriminant function 1 represents the inter-call duration (ID), call duration (CD) and number of pulses (NP); function 2 is loaded by dominant frequency (DF). (B) Discriminant functions of advertisement call attributes of *Hyla eximia* (H. e.), *Hyla arenicolor* (H. a.) and *Tlalocohyla smithii* (T. s.) separated according to basin. Discriminant function 1 is largely loaded by ID, CD and NP, and function 2 by DF. The dashed line encompasses the canonical space occupied by calls of *T. smithii* from Balsas (red) and Magdalena (black) based on the functions generated using the calls of *H. eximia*.

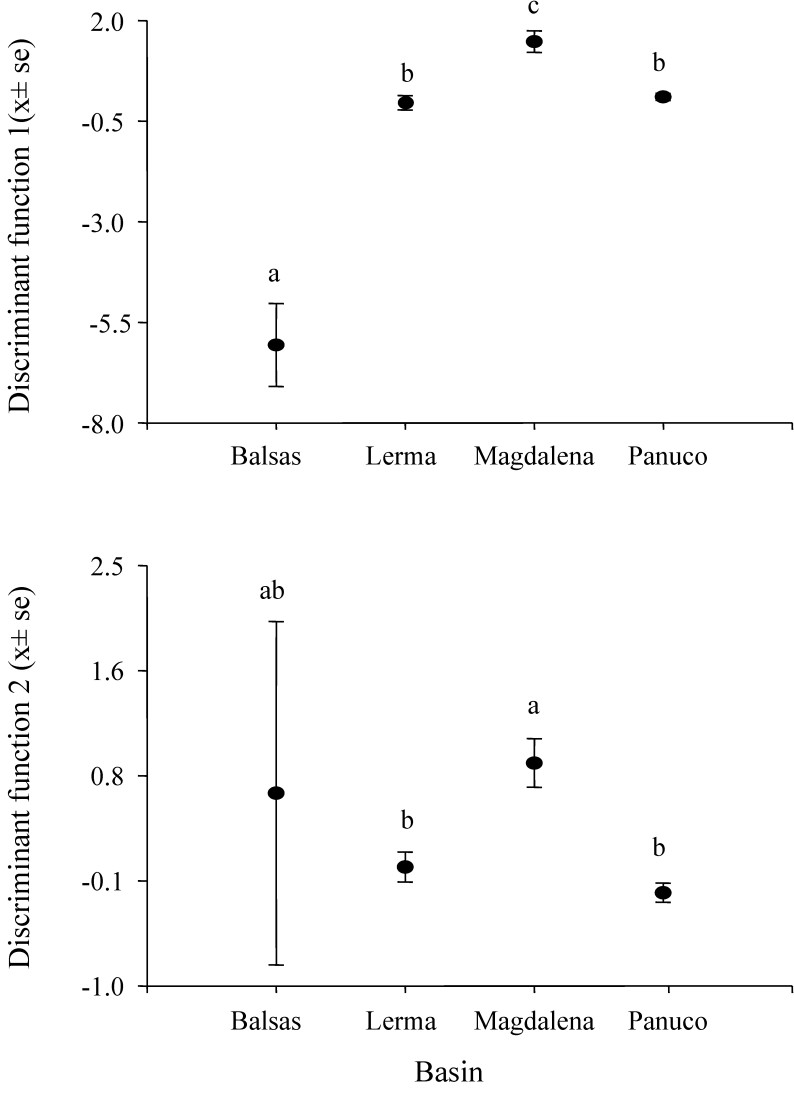

**Figure 4 Scores of *Hyla eximia* calls from the first two canonical functions.** Post hoc tests reveal significant differences (indicated by different letters above the bars) between the mean values of canonical scores of calls from different basins.

*T. smithii* reveals no overlap of canonical acoustic space between the calls of *H. eximia* from the Balsas basin and those of co-occurring *T. smithii*, and a partial overlap between the calls of *H. eximia* at Magdalena those of the local *T. smithii* (Fig. 3). There is complete overlap between the canonical acoustic space of calls of *H. eximia* from the several populations in the Panuco basin, whether they are sympatric with *H. arenicolor* or not, and between those and the calls of *H. eximia* in the Lerma basin (Fig. 3).

## Genetic variation

We found a total of 31 different haplotypes of concatenated mitochondrial genes (Table 4). Nucleotide diversity, number of analyzed samples and total haplotypes per basin for each

**Table 4 Genetic diversity of populations of *H. eximia*.** Haplotype ($h$) and nucleotide ($\pi$) diversity of populations of *H. eximia* from each sub-region of the sampled basins. The two mitochondrial genes were concatenated as well as the two nuclear genes.

| Basin | mtDNA | | | Nuclear | | |
|---|---|---|---|---|---|---|
| | $N, H$ | $h$ | $\pi$ | $N, H$ | $h$ | $\pi$ |
| Northern Panuco | 6, 4 | 0.800 (±0.172) | 0.022 (±0.016) | 5, 4 | 0.900 (±0.161) | 0.123 (±0.080) |
| Central Panuco | 7, 5 | 0.857 (±0.137) | 0.023 (±0.016) | 6, 5 | 0.933 (±0.122) | 0.200 (±0.121) |
| Southern Panuco | 3, 3 | 1.000 (±0.272) | 0.064 (±0.052) | 1, 1 | 1.000 (±0.000) | 0.000 (±0.000) |
| Northern Lerma | 2, 2 | 1.000 (±0.500) | 0.058 (±0.062) | | – | – |
| Central Lerma | 9, 9 | 1.000 (±0.052) | 0.077 (±0.045) | 8, 8 | 1.000 (±0.062) | 0.190 (±0.109) |
| Southern Lerma | 3, 3 | 1.000 (±0.272) | 0.064 (±0.052) | 3, 3 | 1.000 (±0.272) | 0.157 (±0.123) |
| Magdalena | 2, 2 | 1.000 (±0.500) | 0.461 (±0.466) | 3, 3 | 1.000 (±0.272) | 0.118 (±0.094) |
| Balsas | 3, 3 | 1.000 (±0.272) | 0.269 (±0.205) | 2, 2 | 1.000 (±0.500) | 0.088 (±0.095) |

**Notes.**

Standard deviation in parentheses.

$N$, analyzed samples number; $H$, number of haplotypes.

concatenated gene pair are shown in Table 4. Mitochondrial nucleotide diversity was generally low, and the high values from Magdalena ($\pi = 0.462$) and Balsas ($\pi = 0.269$) are probably not reliable since standard deviations are high due to small sample size (Table 4). Mitochondrial haplotype diversity was very high in all regions (Table 4). Since in most cases the number of haplotypes is identical to the sample size, it does not represent the populations' haplotype diversity.

The phylogenetic reconstruction based on mitochondrial genes recovers four major lineages of *H. eximia* loosely grouped by basin, with some admixture (Fig. 5). The basal clade (A) includes frogs from the southern reaches of the Panuco drainage, at the edge of the ancient closed basin of Apan, as well as frogs from the Balsas and one from Magdalena. This is the best -supported clade with bootstrap values of 56–96%. The next mitochondrial clade to diverge (B) includes frogs from central Lerma, Magdalena and Balsas basins. The most recent, sister clades include only Lerma (C) frogs, or frogs from north and central Panuco and central Lerma (D). Bootstrap values are higher in mitochondrial genes than in nuclear genes, and for the basal groups more than for the recent groups. As expected from the above reconstruction, the populations comprising clade A (southern Panuco, Balsas and Magdalena) are the most differentiated; they differ mostly (and significantly) from populations in clade D; central and northern Panuco, as well as from central Lerma as shown by the $F_{ST}$ values (see Table 5).

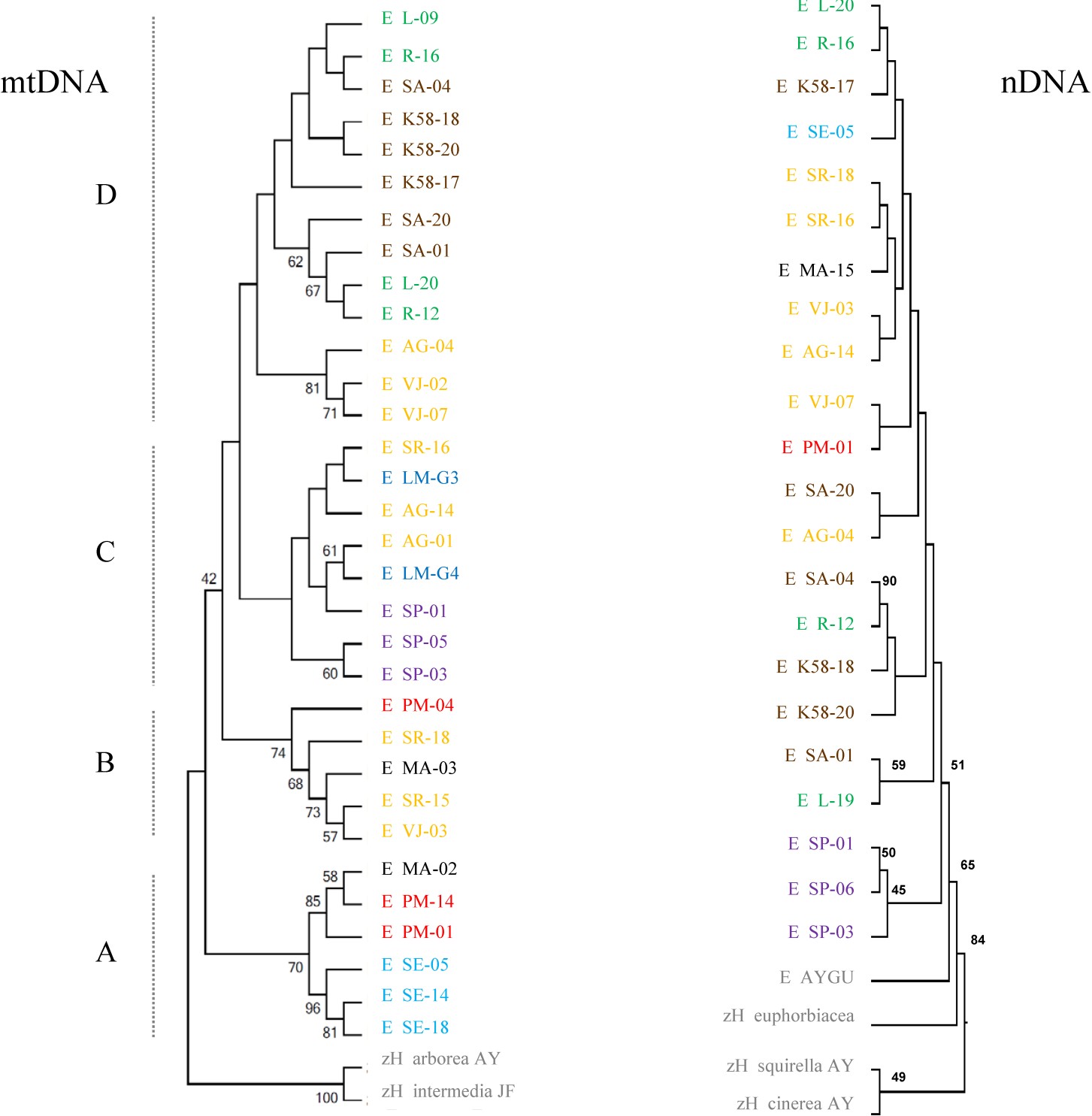

**Figure 5** **Phylogeny of concatenated mitochondrial (ATPase and cyt-*b*) and nuclear (POMC and Rho) genes.** Individuals are separated by basin: northern Panuco (green), central Panuco (brown), southern Panuco (turquoise blue), northern Lerma (blue), central Lerma (yellow), southern Lerma (purple), Magdalena (black), Balsas (red) and out groups (gray). Letters to the left designate proposed clades.

**Table 5 Population differentiation of *H. eximia*.**

| Basin | Northern panuco | Central panuco | Southern panuco | Central lerma | Southern lerma | Magdalena | Balsas |
|---|---|---|---|---|---|---|---|
| Northern Panuco | – | −0.025 (P = 0.513) | 0.133 (P = 0.405) | **0.175** (P = **0.009**) | **0.304** (P = **0.036**) | **0.303** (P = **0.045**) | 0.300 (P = 0.054) |
| Central Panuco | −0.088 (P = 0.639) | – | 0.189 (P = 0.117) | 0.060 (P = 0.090) | **0.239** (P = **0.000**) | 0.089 (P = 0.234) | 0.075 (P = 0.225) |
| Southern Panuco | **0.904** (P = **0.027**) | **0.915** (**P = 0.018**) | – | **0.182** (P = **0.045**) | 0.223 (P = 0.486) | 0.294 (P = 0.099) | 0.363 (P = 0.333) |
| Central Lerma | 0.101 (P = 0.090) | 0.077 (P = 0.126) | **0.840** (P = **0.000**) | – | **0.236** (P = **0.027**) | −0.017 (P = 0.504) | 0.098 (P = 0.063) |
| Southern Lerma | **0.328** (P = **0.027**) | **0.309** (P = **0.009**) | 0.863 (P = 0.090) | 0.131 (P = 0.063) | – | 0.268 (P = 0.063) | 0.299 (P = 0.18) |
| Magdalena | 0.334 (P = 0.081) | 0.445 (P = 0.054) | 0.565 (P = 0.099) | 0.364 (P = 0.387) | 0.161 (P = 0.108) | – | −0.031 (P = 0.396) |
| Balsas | **0.538** (P = **0.018**) | **0.611** (P = **0.000**) | 0.628 (P = 0.153) | **0.560** (P = **0.000**) | 0.409 (P = 0.198) | −0.431 (P = 0.495) | – |

**Notes.**

Mitochondrial $F_{ST}$ below, and nuclear $F_{ST}$ above the diagonal.

The mitochondrial haplotype network seems to reflect the distribution of individuals into the major basins (Fig. 6). Again, haplotypes from southern Panuco are not connected to any other basin. Two of the three Balsas haplotypes are also isolated from those from other basins, except for one haplotype from Magdalena (the other Balsas haplotype is linked to haplotypes from central Lerma). By and large, the haplotypes from central and northern Panuco are closely linked, and relatively closely linked also to those from the three Lerma sub basins (Fig. 6).

We found 25 different haplotypes of concatenated nuclear genes (Table 4). The greatest nucleotide diversity of nuclear genes was found in central Panuco ($\pi = 0.200$) central ($\pi = 0.190$) and southern Lerma (0.157), but the latter has a high standard deviation (Table 4). Just as with mitochondrial genes, nuclear haplotype diversities were often $\simeq 1$ and are certainly overestimations.

The reconstruction based on nuclear genes is much less clear, with very shallow branches, although it also suggests a link between Lerma and both Balsas and Panuco (Fig. 5). The degree of nuclear genetic differentiation of *H. eximia* between regions is roughly consistent with that shown by mitochondrial genes: populations from southern Lerma (mitochondrial clade A) are significantly differentiated from those in northern and central Panuco, as well as central Lerma (mitochondrial clade D), yet this latter clade is formed by frogs from regions that are significantly differentiated by nuclear genes. Frogs from Magdalena (mitochondrial clade B) are significantly differentiated from those from northern Panuco (mitochondrial clade D; Table 5).

The Evanno method obtained a true value of $K = 3$ which is, however, not too sharply delimited. Both genes show different patterns among the populations analyzed and little correspondence to the patterns inferred from mitochondrial genes (Fig. 7). This could be due to recombination or to the fact that Structure fails to find differentiation when

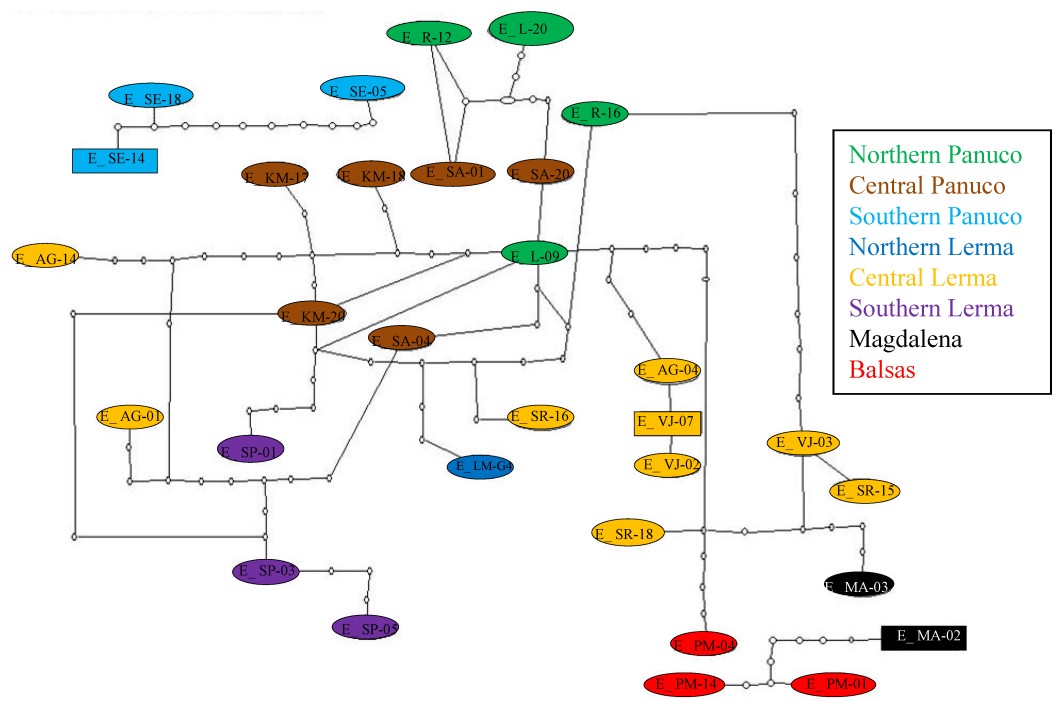

**Figure 6 Haplotype network of concatenated mitochondrial genes.** Haplotypes correspond to each of the basins' sub-regions.

the population structure is too subtle (*Latch et al., 2006*), which could be our case as the reduced sample size failed to produce significant values of population differentiation.

### Phenotypic, genetic and geographical distance

There was no significant correlation between call variation and geographic distance (Mantel test $r = 0.125$, $P = 0.306$) even when controlling for mitochondrial (Mantel partial test $r = 0.126$, $P = 0.218$) or nuclear ($r = 0.185$, $P = 0.234$), $D_{xy}$. Nuclear $D_{xy}$ was not correlated with geographic distance ($r = 0.097$, $P = 0.311$). This pattern remains when controlling for call variation ($r = 0.167$, $P = 0.293$). We did find a significant correlation between geographic distance and mitochondrial $D_{xy}$($r = 0.372$, $P = 0.035$), and this association remained when controlling for call variation ($r = 0.372$, $P = 0.042$). Call variation was not correlated with either mitochondrial ($r = 0.022$, $P = 0.409$) or nuclear ($r = 0.426$, $P = 0.108$), $D_{xy}$, even when controlling for geographic distance (mitochondrial $D_{xy}$, $r = 0.026$, $P = 0.419$; nuclear $D_{xy}$, $r = 0.443$, $P = 0.128$).

## DISCUSSION

### Song description

Intraspecific variation in the attributes of the advertisement calls of *H. arenicolor* and of the *H. eximia* species group is common (e.g., *Blair, 1960*; *Klymus et al., 2010*), but its origin and biological significance are not fully understood. A comprehensive study of call variation

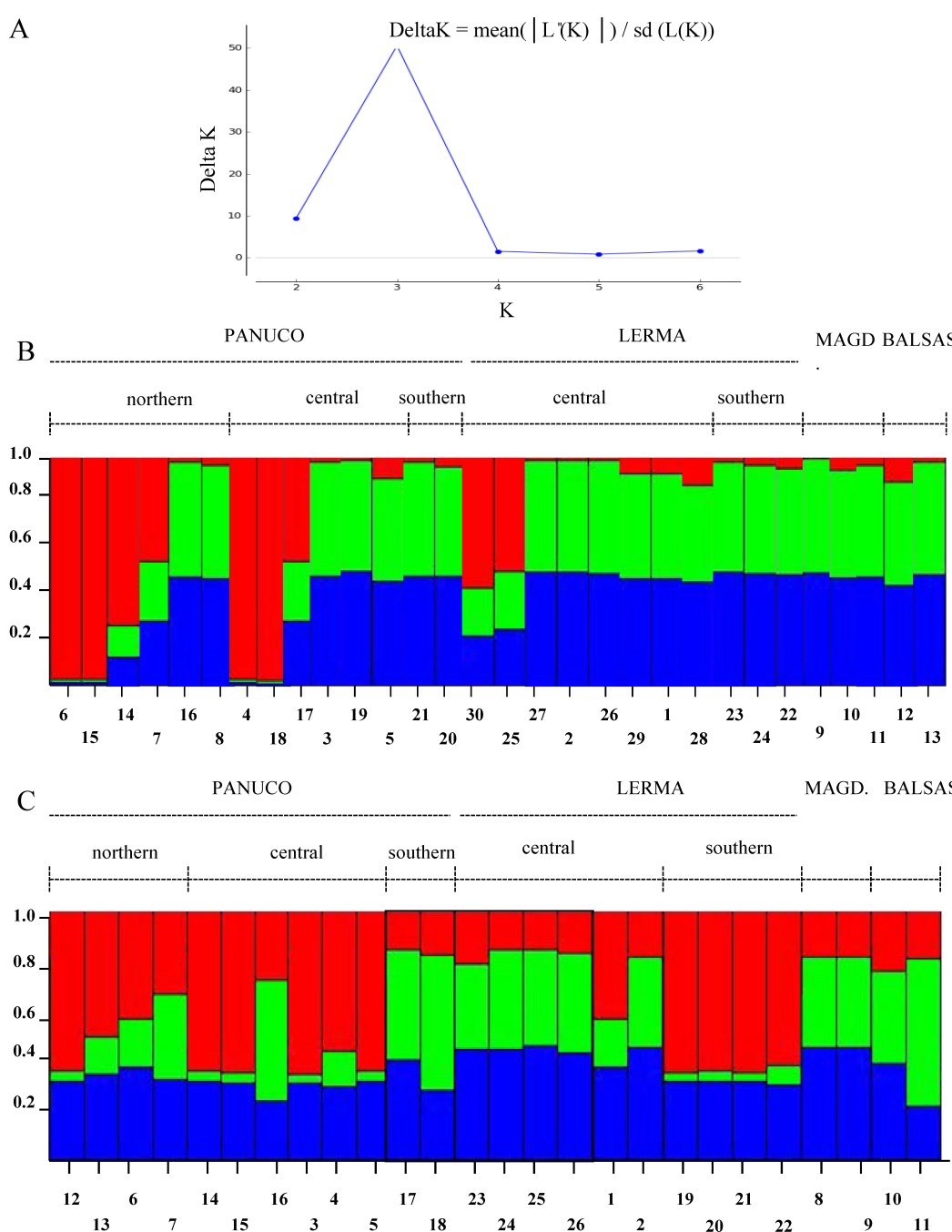

**Figure 7 Population structure of sampled individuals of *H. eximia*.** (A) Plot of Delta *K* (change in the probability of *K*) as a function of *K* (=number of groups). (B) Population structure infered using POMC nuclear gene. (C) Population structure inferred using Rho nuclear gene. Barplots in B and C show the likelihood (Y-axis) of each individual's (X-axis) assignment to a particular population for *K* = 3, determined with the Evanno method.

across the whole geographic range of the Canyon treefrog revealed that such differences are not large enough to promote speciation, although a degree of assortative mating preferences was evident over large geographic scales (*Klymus & Gerhardt, 2012*), and at least some of that variation may be linked to introgression with species in the *H. eximia* group (see *Klymus et al., 2010*). Three populations of *Hyla wrightorum*, a member of the latter group, were found to produce calls of different dominant frequency (and two differed in the duration of their calls; *Gergus, Reeder & Sullivan, 2004*). Pulse rate, an attribute often involved in mate recognition in these treefrogs (e.g., *Klymus & Gerhardt, 2012*), was not different between populations, thus the authors suggested that call variation in *H. wrightorum* is unlikely to promote mating isolation (although we note that differences in dominant frequency may be due to local adaptation to facilitate transmission (*Littlejohn, 1970*) and could thus lead to assortative mating).

Variation in the calls of *H. eximia* has been less comprehensively studied. An early report by *Blair (1960)* classified some populations as either producing "slow" (PR ~50 Hz) or "fast" (PR ~100 Hz) calls. That study was not concerned with exploring the possible origin of such variation but we now know that two populations then categorized as producing slow calls correspond to chorus where *H. eximia* is sympatric with *T. smithii*; in these we recorded songs with low PR and relatively high PPF and DF. In fact, the calls of frogs recorded in sympatry with *T. smithii* were more variable (mean CV = 29%) than those recorded elsewhere (see Table 2 and Fig. 2). This is unlikely to be an artifact of a small sample size, as this estimate is based on a larger sample than that used to assess variation in the attributes of calls produced in sympatry with *H. arenicolor*. In mixed choruses, calls of *H. eximia* may be masked by the long and/or the short notes of *T. smithii* calls. This may mean that males of *H. eximia* confront unpredictable note periodicity, since calls of *T. smithii* are made of different notes, both of variable length, and each with a different and also variable period (Table 3). This would promote high variation of inter-call interval in *H. eximia*, which may explain the high coefficient of variation of IC in those sympatric localities (Table 2).

We could not discern a link between type of variable (temporal, frequency, structural) and the amount of variation it harbors (Table 2; Fig. 2), but pulse peak frequency, dominant frequency, call duration and pulse duration were the least variable elements of *H. eximia* calls, and may thus be useful elements for species recognition.

## Call variation

In his comprehensive work, *Duellman (1970)* and *Duellman (2001)* suggested that *H. eximia* lacks a typical call. This was puzzling since anuran advertisement calls are often stereotypical, which is why they often constitute pre-mating barriers between closely-related species. We did find a substantial degree of variation in calls, but mostly restricted to two geographic regions: Balsas and Magdalena (Fig. 3). Calls of *H. eximia* from these two basins were significantly different from calls elsewhere, mostly in structure/length (IC, CD, NP; discriminant function 1). Number of pulses (NP) was highly correlated with pulse rate (PR; e.g., *H. eximia* in contact with *T. smithii*: 0.9204), which together with dominant

frequency plays an important role in species recognition (*Littlejohn, 1965*; *Gerhardt & Davis, 1988*; *Gerhardt, 1994*) and mate discrimination in other hylids (e.g., *Dendropsophus ebraccatus (Hyla ebraccata)*, *Wollerman, 1998*). There are no published studies of mate choice in *H. eximia*, but in the related *H. arenicolor* female mate choice is influenced by call PR (*Klymus & Gerhardt, 2012*), which is a key element for maintaining isolation between lineages of *H. arenicolor* and might also lead, through its correlation with NP, to reproductive isolation between populations of *H. eximia*. Call (CD) and inter-call duration (IC) reflect calling effort, and the energy required to increase them depends on body size (*Gerhardt, 1994*). We found no differences between populations or basins in body size, thus it is possible that variation of call structure in the Balsas localities is adaptive, enabling *H. eximia* to avoid interference from the calls of *T. smithii* (see below).

Calls of *H. eximia* in Balsas and Magdalena were also different in frequency (DF; function 2) from calls elsewhere. This is a static trait (*Gerhardt, 1991*) that, like other such attributes of treefrog calls, is determined by morphology (several body structures, muscle size, larynx structure, among others; *Dubley & Rand, 1991*; *McClelland, Wilczynski & Ryan, 1996*; *Wells, 2001*; *Gerhardt & Huber, 2002*), resulting in structural similarity of notes and seemingly simple calls (compared to the more complex bird song). A change in this attribute would be energetically costly, and it is therefore likely that the observed variation in DF is the result of selection to improve communication efficiency, which may include avoiding interference from heterospecific calls.

In the Balsas and Magdalena basins, *H. eximia* shares choruses with *T. smithii*. Calls of *H. eximia* from Balsas are very variable, and their canonical acoustic space (as defined by discriminant functions 1 and 2; Fig. 3) is different from that of the local *T. smithii*, which seems to indicate reproductive character displacement, a possibility that requires corroboration from more intense sampling and phonotaxis studies. Interestingly, the distribution in the same canonical space of the calls of *H. eximia* from Magdalena shows that they are shifted in the opposite canonical direction than in Balsas. This can be caused by random differences being selected in different populations, as long as they reduce overlap with *T. smithii*. However, we note that in Magdalena *H. eximia* also faces interference with another frog species; *Gastrophryne sp*. Probably because this is a larger frog, its calls are much longer and lower-pitched (*Duellman, 2001*) than to those of *H. eximia*. This results in a more limited acoustic space available for *H. eximia* to accommodate its call to avoid masking, being simultaneously limited by the temporal and spectral attributes of the calls of *T. smithii* and of *Gastrophryne sp*.

While at Balsas and Magdalena the calls of *H. eximia* differ significantly from those of their conspecifics elsewhere (and are completely different from those of *T. smithii* in Balsas), calls from localities sympatric with *H. arenicolor* (mostly in the Panuco basin) were similar to those from allopatric populations. This was unexpected since, although both species are morphologically very different (*H. arenicolor* has a rough, granulated skin and is somewhat larger than smooth-skinned *H. eximia*, which is more similar to the smaller *T. smithii*), they belong to the same phylogenetic group, share the same mating ponds, and in the Balsas (where we did not sample mixed choruses of these two species) there

is evidence suggesting ancient mitochondrial introgression (*Bryson et al., 2010*; *Klymus & Gerhardt, 2012*), thus mating interference between them was expected. It may be that, since the calls of *H. eximia* and *H. arenicolor* are substantially different, there is no, or little, interference between their calls to drive character displacement, although it may also be that the interaction of the two species is too recent in northern and central Panuco (see below).

## Genetic variation

While call variation between some regions covered in our sample is supported by mitochondrial phylogeny and population structure, it cannot be properly interpreted as a consequence of the phylogeny of *H. eximia*. The mitochondrial reconstruction is generally well supported, although it draws a puzzling phylogenetic history of *H. eximia*. It suggests, very preliminarily, that the populations in our sample originated in the south-east of the current species distribution, and expanded clockwise in their colonization of what are now the basins of rivers Balsas, Magdalena (Ameca), and central (lower) Lerma, from where they may have reached the northern (Santiago) and southern (upper) Lerma, as well as the central and upper Panuco basins (Fig. 1). Published phylogeographic hypotheses for *H. arenicolor* often include a few sequences of *H. eximia* and cover a limited portion of its geographic range, so that no phylogeographic hypotheses for this species can be drawn from them (see *Bryson et al., 2010*; *Klymus & Gerhardt, 2012*). Thus we provisionally propose the above phylogenetic scenario of *H. eximia* based on our best-resolved (mitochondrial) reconstruction (Fig. 1).

## Phenotypic, genetic and geographical distance

Call variation between the regions covered in our sample may not be random (i.e., due to drift); it was greater in particular basins (Balsas and Magdalena) and chorus compositions (*H. eximia–T. smithii*), but did not correlate with genetic or geographic distance. Absence of a correlation between acoustic and genetic distance has been reported in other studies. For instance, in *Oophaga* (*Dendrobates*) *pumilio* (*Pröhl et al., 2007*), the correlation is lost when the analysis controls for geographic distance, and a similar effect occurred in some populations of the Túngara frog *Engystomops* (*Physalaemus*) *pustulosus* (*Ryan, Rand & Weigt, 1996*). These findings are perhaps not surprising, since one of the forces that most effectively shape anuran call attributes is female mate choice (*Boul et al., 2007*), and this is most effective when mistakes are penalized as in a hybrid zone. Thus we should in fact expect greater call variation between populations when reproductive character displacement is favored in at least one of them (*Pfennig and Pfennig, 2010*), than when populations are distant geographically but do not face mating call interference. Indeed, we rule out the possibility that a correlation between genetic and acoustic distance exists but was undetected, since we have adequate call samples, and, as expected, mitochondrial genetic distance was correlated with geographic distance, making it likely that our genetic screening is unbiased.

We have argued that call variation in the regions where *H. eximia* and *T. smithii* coexist is consistent with the expected effects of acoustic interference, but it might also be the

result of morphological differentiation in attributes that influence call production (e.g., the body size in *Hyla leucophyllata*, *Lougheed et al., 2006*; see also *Emerson, 2001*; *Gerhardt & Huber, 2002*; *Castellano et al., 2002*; *Hoskin et al., 2005*). However, an exploratory analysis failed to detect significant differences in body size among populations of *H. eximia*, thus it is unlikely that the variation in calls found between basins is due to differences in the body size. Other causes of call differentiation include local differences in call assemblage (*Wollerman & Wiley, 2002*), in signal transmission efficiency (related to the type of habitat), or in sexual selection (acting through reproductive isolation mechanisms). The first is compatible with our explanation of call differentiation to avoid masking by the calls of *T. smithii*, whereas the reported lack of differentiation amongst the populations not sympatric with *T. smithii*, which cover a much larger and presumably environmentally variable geographic area, argues against the other two. We thus provisionally propose that the geographic variation of advertisement calls of *H. eximia* in the Balsas and Magdalena basins populations is due to reproductive character displacement produced by sharing mating choruses with the related and morphologically similar *T. smithii* and with the distantly related *Gastrophryne sp.*

## ACKNOWLEDGEMENTS

Field work was conducted under SEMARNAT license SGPA/DGVS/03580/09, and was made possible by the help of J Ávila, A Archundia, A Briones, J Carillo, C Chávez, MA Martínez, M Nuñez, Tomás, MJ Godínez, B Peña, I Barbosa, C Montes, N Lifshitz, E Quiróz, N Miranda, M Suárez, M Méndez, E Bermúdez, R Beamonte, F Mendieta, A Freyermuth, C Ríos, A García, E Vázquez and E Ávila-Luna, who also provided logistical support. G Cortés-Soto and A Ríos-Chelén helped with the initial sound analyses, S Benitez-Vieyra and a former referee provided useful statistical advice and L Kiere helped improving the English. This project was founded with a grant from CONACyT (83779) to CMG, and a scholarship to RERT also from CONACyT. This paper constitutes RERT's partial fulfilment of the Graduate Program in Biological Sciences of the National Autonomous University of Mexico (UNAM).

### Funding

This work was financed with funds from a CONACyT (Consejo Nacional de Ciencia y Tecnología; the Mexican Science Council) grant 83779 to CMG, and a scholarship to RERT from CONACyT. The funders had no role in study design, data collection and analysis, decision to publish, or preparation of the manuscript.

### Grant Disclosures

The following grant information was disclosed by the authors:
CONACyT (Consejo Nacional de Ciencia y Tecnología; the Mexican Science Council): 83779.

## Competing Interests

The authors declare there are no competing interests.

## Author Contributions

- Ruth E. Rodríguez-Tejeda conceived and designed the experiments, performed the experiments, analyzed the data, wrote the paper, prepared figures and/or tables, reviewed drafts of the paper.
- María Guadalupe Méndez-Cárdenas performed the experiments, analyzed the data, prepared figures and/or tables, reviewed drafts of the paper.
- Valentina Islas-Villanueva analyzed the data, prepared figures and/or tables, reviewed drafts of the paper.
- Constantino Macías Garcia conceived and designed the experiments, contributed reagents/materials/analysis tools, wrote the paper, prepared figures and/or tables, reviewed drafts of the paper.

## Animal Ethics

The following information was supplied relating to ethical approvals (i.e., approving body and any reference numbers):

The project was approved by the Consejo Técnico de la Investigación Científica (CTIC), the Science Council of our university (UNAM). There is no specific ethical committee in our university for reviewing proposals of this type (the ethical committees deal mostly with medical proposals and with the ethics of academic interactions), but all projects have to be approved by the internal council of scientific research which looks at, amongst other things, the ethical aspects of all projects.

## Field Study Permissions

The following information was supplied relating to field study approvals (i.e., approving body and any reference numbers):

Secretaría de Manejo y Aprovechamiento de los Recursos Naturales (SEMARNAT)–the Mexican Ministry for thew Environment–provided the licence to conduct field work on the species included in this report, and to collect tissue samples for DNA extraction (SGPA/DGVS/03580/09).

## DNA Deposition

The following information was supplied regarding the deposition of DNA sequences:

We have submitted the sequences to GenBank and have accession numbers for the nuclear and mitochondrial genes; these can be found in Table S2 (GenBank submission ID 1700625).

## Supplemental Information

Supplemental information for this article can be found online at http://dx.doi.org/10.7717/peerj.420.

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
