# Peer review of "Geographic variation in the advertisement calls of Hyla eximia and its possible explanations"

_PeerJ, doi:10.7717/peerj.420_

## Round 0.1 · original submission · Major Revisions

Overall, I liked the study which takes an interesting approach to an old problem with a relevant group of organisms. However, both reviewers have significant misgivings about the validity of your conclusion that reproductive character displacement has been identified in populations of H. eximia and T. smithii. One reviewer suggests that you reformulate your hypothesis so that this is not the focus of the paper, but a possible finding worthy of further investigation, and I concur that this would appear sensible given the data available in this manuscript, and this is the reason I have chosen a decision of Major Revision.

I have the following additional observations which I would like addressed in any future submission:
- The abstract overly simplistic and should be reformulated including an excerpt of relevant results. Currently, it appears almost to be written as an afterthought.
- The manuscript is peppered with typos and grammatical errors which detract from the generally good quality of the writing.
- Given the very wide range of Hyla eximia, and your relatively limited sampling, I’m uncomfortable with your discussion of phylogeographic history (lines 411-444). One reviewer also has similar comments regarding your suggested discussion of dates. With more comprehensive sampling and analyses this would be interesting, but it is currently beyond the scope of the data you present.
- Line 74-75: "More recently, based on distributional, phonetic and genetic data G. Cortés-Soto et al. (unpublished data)" I'm uncomfortable with this statement as if the data is unpublished, then the statement hasn't actually been made. Unpublished data is just that, unpublished statements are not the same. Moreover, I find this pseudo-citation slightly disturbing as the subject matter is clearly the same as your manuscript (H. eximia, acoustic and genetic data), G. Cortés-Soto is not a co-author and not mentioned in the acknowledgements (the et al are never mentioned - perhaps that includes you or some of your co-authors?). You will need to either clarify this statement with a citation or change it to a personal communication following the PeerJ guidelines on this.
- Line 196-7: Does this relate to ethics approval? If so, please state.
- Many of our readers won’t be familiar with Mexico and it would be good to have a map showing all sampling localities. Perhaps this could be colour coded as with Figure 6 and include sample sizes?
- I’d suggest combining Figures 7 & 8 and showing better the three divisions found. Although Structure found K=3, the figures aren’t very convincing of this finding. If K=3 is relevant to the revised objective of the paper, could you please include a plot of K and Ln Pr(X | K)?

Reviewer 1 ·

Basic reporting

See below

Experimental design

See below

Validity of the findings

See below

Additional comments

Review of:
Macias Garcia et al., Can acoustic interference and genetic differentiation resolve the mosaic variation of advertisement calls across the geographic range of the mountain treefrog (Hyla eximia)?


General comments

The authors investigate whether acoustic niche partitioning in sympatry with other related species explains divergence in advertisement calls among populations of the species Hyla eximia.

While the question at hand is interesting, the manuscript needs much improvement both regarding the used methods, as also with regards to Command of English. The paper needs to be corrected by a native English speaker, or via contracting an external editing service. Most notably, the Methods and Results sections need to be improved. Third, before the manuscript was sent out for review, it should have been checked for compliance with the journal style guidelines. The legends of all figures and legends don’t correspond to each other (an error caused by the Submission system?), and thus it is difficult to understand the manuscript.

The results aren’t significant, which I assume is due to inappropriate statistical tests, nonetheless the discussion is very long and built on speculations.
The statistical tests used are partially problematic. For example, the authors test matrix correlation among call, genetic, and geographic datasets using partial Mantel tests. According to several recent papers (including Legendre & Fortin, 2010), the partial Mantel test should not be used for comparisons involving a spatial component. Instead, they should look into ordination-based methods (e.g., redundancy analyses). All these methods are publicly available via the R platform. Another alternative would be spatially corrected regression (Moran’s Eigenvector, in ArcGIS). This part of the manuscript needs to be re-worked in order to yield more solid results. As a help, I recommend these publications:

http://onlinelibrary.wiley.com/doi/10.1111/j.1755-0998.2010.02866.x/abstract
http://www.scielo.br/scielo.php?pid=S1415-47572013000400002&script=sci_arttext
http://www.academia.edu/730273/Alternatives_to_the_partial_Mantel_test_in_the_study_of_environmental_factors_shaping_human_morphological_variation

Secondly, the authors test difference in calls among sites/populations that are sympatric or allopatric with other species would be to code sympatry vs. allopatry as a categorical variable to explain overall call variance. To require significant results from such an analysis, this requires the evolution of call divergence to be equidirectional in all instances of sympatry (i.e., producing a common mean in call variables versus allopatric calls), which is unlikely to be the case since the other hylid species have calls that are very different from each other and thus provide a very variable acoustic environment among sympatric sites (i.e., shifting the mean to a value between two clouds of data points). This partially might explain the insignificant results from these tests.

An alternative way of testing acoustic niche partitioning within the species would be to compare call divergence from sympatric Hyla eximia to the respective sympatric species, versus allopatric populations from Hyla eximia. If acoustic niche partitioning is the driver for H. eximia call divergence, one would expect to find less overlap in H. eximia calls to the calls of the other species in these populations, that also might or might not diverge from allopatric populations of Hyla eximia.
A general caveat for the sympatry/allopatry scenario is that in theory (and often in practice), acoustic niche partitioning can follow the temporal dimension: frogs from different species would alternate choruses temporally so no call divergence is necessary.

Another caveat that I would like to be discussed more prominently is that the dominant frequency, which provides the major axis of intraspecific call divergence, is known to vary according to the environment (its signaling transmission properties). Divergent calls might simply be a result of more or less dense calling environments. If some data on these sites is available (if only coded categorically), this might be worthwile to investigate.

The phylogeny provides almost no resolution, as well the results of STRUCTURE clearly show that there is not enough genetic structure among the populations to warrant a priori categorization in different groups belonging to different basins. Another problem (Table 3) is, as the authors also state themselves, that the sampling for the genetic analyses was too small. For studying populations, a minimum number of 15 samples might be necessary. Almost each of the samples from each population had a different haplotype, indicating that the overall haplotype diversity from each population might be much higher, and genetic divergence among populations in form of Fst values should not be estimated from the sample present (see Table 4 or 2)

In summary, this manuscript seems not to be ready for submission yet. I propose that the authors reanalyze their data, streamline the results they want to present (too many figures and tables!), and formulate the expectations and significance of their results more clearly, while paying attention to manuscript formatting and readability of the manuscript. .

More specific comments:

L287 and elsewhere: please replace all mentions of “Mt” with the standard acronym mtDNA.
L293 and elsewhere: “Nuclear DNA” there are random subheaders present in the manuscript. I am sure this does not correspond to the formatting guidelines.
L394 and elsewhere: “The apparent displacement in Magdalena runs in the opposite direction of the acoustic space” This is simply not appropriate English.
L495 “Call divergence between the regions covered in our sample may not be random (…) it was supported by phylogeny (...) but did not correlate with genetic distance” This sentence is contradictory and a good example for the amount of speculation present in the manuscript.

Figures and Tables

Table 1. Table headers don’t correspond to columns. Why are three values missing?
Table 2. Wrong caption. Shows results from DFA, not from partial Mantel test.
Table 3. If the mitochondrial and nuclear genes were concatenated, why are haplotype and nucleotide diversity shown separately?
Table 4. Shown as table 2
Table 5. belongs to legend of table 2. Here you can clearly see that the information extracted from nuclear genes does not contribute to the overall measured effect.
Figure 1. What do the symbols on the right set of images belong to? Which units are the Y-axes in? I tried to find this out by relating the dominant frequency of H. eximia (98Hz) of Table 1 to this figure but unsuccessfully so (values here are around 3KHz).
Figures 2,3: Looking at these I find it hard to believe that the DFA scored 57% of these calls correctly according to basin of origin.
Figure 4. “Significant differences are indicated by different letters above the bars”. Differences to what? These are mean-with-error plots, I don’t understand which test of difference these correspond to. And what exactly do the letters a, b, c, and ab tell the reader about significance?
Figure 5: The correct abbreviation for cytochrome b gene is cob not cytB, the correct abbreviation for the rhodopsin gene is Rhod. None of the clades are statistically supported, so a bifurcating phylogeny is not an appropriate way to depict relationships among the individuals (also see Figure 6). A specimen table with locality information is missing. Genbank numbers need to be obtained (instead of providing an alignment in text format as In Appendix A).
Figure 7 & 8. “Maximum Likelihood (y-axis) was reached when the average K=3 groups” ?????

Figure “Phylogeographic Hypotheses”: The allopatric populations are indicated by an image of the frog that on the map looks either way, whereas in the legend it faces the right, and the sympatric populations face left. The figure would be better understandable if the allopatric frog symbols would all face to the right side.

Reviewer 2 ·

Basic reporting

This manuscript reports the geographic variations of advertisement calls and genetic divergences in three closely related frog species, Hyla eximia, H. arenicolor, and Tlalocohyla smithii. Specifically, the authors tried to understand the evolution of call variations by testing the reproductive character displacement (RCD) and isolation by distance (IBD) hypotheses. The main findings are that the differences in call properties among basins were congruent with the phylogenetic patterns and the reproductive interaction with T. Smithii might have shaped the call variation of H. eximia. The extent of background information is adequate, and the scope of this research well fits into the journal. The submission is self-contained and includes all results relevant to the hypotheses.

Experimental design

No comments

Validity of the findings

The major problem in this manuscript lies in the sample size to address the RCD hypothesis. The call properties can be influenced by many unknown environmental factors in a population. When populations of two or more species are compared for call properties, it's difficult to identify selective factors for call variations . Even if there are differences in call properties in two species, the differences may arise from solely environmental factors. Thus, sampling should be conducted in many localities across a wide range of environmental conditions (See Gabor & Ryan. 2001. Geographic variation in reproductive character displacement in mate choice by male sailfin mollies. Proceedings of the Royal Society B. 268: 1063-1070). I don't think that the sampling in this study is adequate to address the RCD hypothesis.

However, the sampling problem may be remedied by shifting the goal of this manuscript a little bit. Given the samples in this manuscript, authors may adequately test whether IBD or geographic barriers are critical for patterns of call variation and genetic divergence in H. eximia (Prohl et al. 2006. ; Jang et al. 2011. Geographic variation in advertisement calls in a tree frog species: Gene flow and selection hypotheses. PLoS ONE 6: e23297). When the reanalysis is attempted, the unit of comparison should be populations, not individual frogs. The call properties of individual frogs should be averaged for each population, then population means should be used to test hypotheses.

L59: phylopatric? Is it philopatric?
L60: Genetic structure of a population is not high or low. It is typically described as simple or complex, highly differentiated or not.
L75: What is masl?
L88: Specify the public databases?
L86-93: Study localities are not clearly identified in M&M. Include the following information in the distribution map.
• Geographic distributions of Hyla eximia, H. arenicolor, and Tlalocohyla smithii.
• Recording localities
• Explicit patric relationships among three frog species.

Although there is only one species found in a locality, this locality may not be allopatric in relation to other species. Amphibian populations may suffer from local extinction in a locality, adjacent populations may recolonize the locality following years. Furthermore, amphibian populations decline rapidly around the world. Thus, information about current distributions as well as historic distributions are necessary to evaluate the patric relationships among species (See Park et al. 2013. No reproductive character displacement in male advertisement signals of Hyla japonica in relation to the sympatric H. suweonensis. Behavioral Ecology and Sociobiology 67:1345-1355).

L107-111: Define call characteristics. What is the difference between dominant frequency and fundamental frequency?
L124: Chorus type is misleading. Use patry (sympatry vs. allopatry) instead.
L203-211: It's unclear whether values of call characters were adjusted for temperatures in Table 1. Adjusted values should be represented to make comparisons easy across the species.
L237-241: Instead of eliminating some call characters over others arbitrarily, use principal component analysis to reduce call dimensions for further statistical analyses.
L260-267: As pointed out earlier, it's difficult to draw a conclusion about interspecific reproductive interaction as a selective force for call variation, based on a few locality data.

Additional comments

No comments

Reviewer 3 ·

Basic reporting

1. Abstract.
The use of "acoustic space" borders on jargon and should be changed for more explicit terms or explained.
The phrase "call divergence" is ambiguous here; diverging from what?
The phrase "behavioral divergence" is used as "call divergence", such a shortcut is unjustified here.

2. Text
l. 44. Authors generalized chorus calling to all anurans, which is far from being the case. Rephrase to make the statement less general.
l. 51. precise what may mask male calls.
l. 52. explicit what the "acoustic space" is.
l. 58. again, authors generalized and make all anurans aquatic obligates, which is not the case.
l. 95. The authors mention 177 advertisement calls recorded, whereas at l.200 only 169 are used for analyses. The number of bad quality recordings is not relevant in the paper.
l. 100. Explain how the body temperature was measured. Was it a probe? How did the authors ensured minimal heat transfer between the manipulator and the frog?
l. 121. Statistical analyses are only very briefly explained, it needs to be expanded and the methods used better justified.
l. 123. Precise after mentioning analysis of covariance : “(ANCOVA)”
l. 128. Explicit the meaning of MANOVA for its first use.
l. 130. Precise what hypothesis and why a non-significant nesting is expected.
l. 139. Change “mtDNA” for “mitochondrial”
l. 150. Same comment as l. 139.
l. 167. “a” missing in second word.
L. 171-173. Give the full names first and the acronyms after and in brackets.
l. 175. Give full name for AIC
l. 277. Replace “phylogeographic” by “phylogenetic” the reconstruction did not include geographic information, but only genetic information. The geographic information is only represented by the colors of the tree tips.
l. 285. “They differ” in what? Is it call variables? Explicit.
l. 291. Replace “by en large” by “by and large” which is the correct form.
l. 307-308. Sentence is confusing, rephrase to make clearer.
l. 362. Replace “eximia “ by “H. eximia”
l. 363. “y” missing on last word.
l. 381-387. Paragraph very confusing. Rephrase to explicit better.
l. 394. What does “run in the opposite direction of the acoustic space” mean? Rephrase to explicit better.
l. 396. I believe it is the first mention of Gastrophryne sp. Clarify the idea maybe by expanding the role it could play.
l. 405. Explicit the use of “introgression” (of what? In what?)
l. 407. Replace “too” by “very” or similar.
l. 448. Dendrobates pumilio is not a hylid frog, the use of “For instance” at the beginning of the sentence implies it is. Also I suggest the use of the taxonomy from Frost (2013- http://research.amnh.org/vz/herpetology/amphibia/): Oophaga pumilio.
l. 450. Similar to l. 448: Frost (2013) taxonomy recognizes the genus Engystomops for the Tungara frog.
l. 451-452. The authors are very categorical about sexual selection’s effect on advertisement calls. Please cite references and give a more nuanced statement.
l. 475. The conclusion on Gastrophryne sp. is important although this species has barely being mentioned before. Please expand, maybe also in previous sections.

3.Tables and Figures.
Table 5. Replace “tabla” by “table”.
Figure 4. The legend is too succinct. Make the letter system more explicit.

Experimental design

As previously said, I think the statistical analyses must be further explained and justified, and other types of statistical analyses (such as glmm for example) may have been more appropriate.
l. 127. The residuals are not ensured to be normally distributed; using the residuals is not a guarantee to meet the assumption of normality. Furthermore, the use of residuals can cause problems, see Darlington and Smulders (ANIMAL BEHAVIOUR, 2001, 62, 599–602 ).
l. 275. and l. 294. The haplotype diversity analysis is irrelevant since, as the authors say, the sample size is too small. I would suggest to remove the part altogether.

Validity of the findings

The authors conclude the possibility of RCD for H. eximia sympatric to other hylids. This conclusion is however supported only by one of the two populations in this context. The results and conclusions need to be more detailed and more nuanced, and alternative hypotheses need to be proposed for the observed difference between the to populations of H. eximia.

l. 255. The authors say that Balsas and Magdalena populations of H. eximia are distinct from the others, whereas the population of Magdalena largely overlaps the populations of Panuco and Lema (Fig. 2). I therefore do not understand or agree with the author’s conclusion.
l. 388-390. The authors conclude that there is a RCD between the two frog species, while only one population over two would fit this hypothesis: the two species at Magdalena basin largely overlap (Fig. 3).
l. 421-444. Although hypotheses on the historical phytogeography of H. eximia is very interesting, the authors mention periods of possible colonization without any information on the age of the divergence between the populations. I would suggest dating the phylogeny, which would allow to talk about geographical condition at reasonable periods, or to provide references documenting these divergences ages. Without any divergence time information, I would refrain from inferring possibilities of colonization timing or mechanism.

Additional comments

I think the study is very interesting, with a good problematic. However, much work is needed to improve the clarity of the results, and other statistical analyses may be more relevant. Also dating the phylogeny or citing references where the divergence of these clades is dated would be necessary to talk about colonization scenarios.

---

## Round 0.2 · Minor Revisions

Thank you for your revision which is clearly in line with the reviewers' recommendations, and that some of your more ambitious claims have been reigned in. Although many of the problematic English language uses have been overcome, there are still some that remain. I've highlighted these from the first few pages below:

Abstract
“We investigated whether variation in the advertisement call of the mountain treefrog (Hyla eximia) is linked to geographic distribution in relation to major river basins,…” please include some geographic reference (e.g. country, state, etc).

“…or closely-related species (the dwarf treefrog, Tlalocohyla smithii).” That this species is in another genus suggests that it is not closely-related. Please rephrase. See line 82.

“We found that the multidimensional acoustic space of calls from two basin, those where H. eximia…” Grammar

Line 43-44. As noted previously, this is likely incorrect. There are many other signals not taken into consideration by Wells.

Line 73: “Based on phonetic data (Cortés-Soto, 2003) suggested that H. eximia…” Please correct the citation.

Line 73: masl = m asl.

Line 88: “…same mating pools that H. eximia…” replace “that” with “as”

Please make an effort to examine your text exhaustively and to remove any grammatical problems.

Reviewer 2 ·

Basic reporting

See below

Experimental design

See below

Validity of the findings

See below

Additional comments

General comments

The authors investigate whether acoustic niche partitioning in sympatry with other related species explains divergence in advertisement calls among populations of the species Hyla eximia.

While the question at hand is interesting, the manuscript needs much improvement both regarding the used methods, as also with regards to Command of English. The paper needs to be corrected by a native English speaker, or via contracting an external editing service. Most notably, the Methods and Results sections need to be improved. Third, before the manuscript was sent out for review, it should have been checked for compliance with the journal style guidelines. The legends of all figures and legends don’t correspond to each other (an error caused by the Submission system?), and thus it is difficult to understand the manuscript.

The results aren’t significant, which I assume is due to inappropriate statistical tests, nonetheless the discussion is very long and built on speculations.
The statistical tests used are partially problematic. For example, the authors test matrix correlation among call, genetic, and geographic datasets using partial Mantel tests. According to several recent papers (including Legendre & Fortin, 2010), the partial Mantel test should not be used for comparisons involving a spatial component. Instead, they should look into ordination-based methods (e.g., redundancy analyses). All these methods are publicly available via the R platform. Another alternative would be spatially corrected regression (Moran’s Eigenvector, in ArcGIS). This part of the manuscript needs to be re-worked in order to yield more solid results. As a help, I recommend these publications:
http://onlinelibrary.wiley.com/doi/10.1111/j.1755-0998.2010.02866.x/abstract
http://www.scielo.br/scielo.php?pid=S1415-47572013000400002&script=sci_arttext
http://www.academia.edu/730273/Alternatives_to_the_partial_Mantel_test_in_the_study_of_environmental_factors_shaping_human_morphological_variation

Secondly, the authors test difference in calls among sites/populations that are sympatric or allopatric with other species would be to code sympatry vs. allopatry as a categorical variable to explain overall call variance. To require significant results from such an analysis, this requires the evolution of call divergence to be equidirectional in all instances of sympatry (i.e., producing a common mean in call variables versus allopatric calls), which is unlikely to be the case since the other hylid species have calls that are very different from each other and thus provide a very variable acoustic environment among sympatric sites (i.e., shifting the mean to a value between two clouds of data points). This partially might explain the insignificant results from these tests.

An alternative way of testing acoustic niche partitioning within the species would be to compare call divergence from sympatric Hyla eximia to the respective sympatric species, versus allopatric populations from Hyla eximia. If acoustic niche partitioning is the driver for H. eximia call divergence, one would expect to find less overlap in H. eximia calls to the calls of the other species in these populations, that also might or might not diverge from allopatric populations of Hyla eximia.

A general caveat for the sympatry/allopatry scenario is that in theory (and often in practice), acoustic niche partitioning can follow the temporal dimension: frogs from different species would alternate choruses temporally so no call divergence is necessary.

Another caveat that I would like to be discussed more prominently is that the dominant frequency, which provides the major axis of intraspecific call divergence, is known to vary according to the environment (its signaling transmission properties). Divergent calls might simply be a result of more or less dense calling environments. If some data on these sites is available (if only coded categorically), this might be worthwile to investigate.

The phylogeny provides almost no resolution, as well the results of STRUCTURE clearly show that there is not enough genetic structure among the populations to warrant a priori categorization in different groups belonging to different basins. Another problem (Table 3) is, as the authors also state themselves, that the sampling for the genetic analyses was too small. For studying populations, a minimum number of 15 samples might be necessary. Almost each of the samples from each population had a different haplotype, indicating that the overall haplotype diversity from each population might be much higher, and genetic divergence among populations in form of Fst values should not be estimated from the sample present (see Table 4 or 2)

In summary, this manuscript seems not to be ready for submission yet. I propose that the authors reanalyze their data, streamline the results they want to present (too many figures and tables!), and formulate the expectations and significance of their results more clearly, while paying attention to manuscript formatting and readability of the manuscript.

More specific comments:

L287 and elsewhere: please replace all mentions of “Mt” with the standard acronym mtDNA.
L293 and elsewhere: “Nuclear DNA” there are random subheaders present in the manuscript. I am sure this does not correspond to the formatting guidelines.
L394 and elsewhere: “The apparent displacement in Magdalena runs in the opposite direction of the acoustic space” This is simply not appropriate English.
L495 “Call divergence between the regions covered in our sample may not be random (…) it was supported by phylogeny (...) but did not correlate with genetic distance” This sentence is contradictory and a good example for the amount of speculation present in the manuscript.

Figures and Tables

Table 1. Table headers don’t correspond to columns. Why are three values missing?
Table 2. Wrong caption. Shows results from DFA, not from partial Mantel test.
Table 3. If the mitochondrial and nuclear genes were concatenated, why are haplotype and nucleotide diversity shown separately?
Table 4. Shown as table 2
Table 5. belongs to legend of table 2. Here you can clearly see that the information extracted from nuclear genes does not contribute to the overall measured effect.
Figure 1. What do the symbols on the right set of images belong to? Which units are the Y-axes in? I tried to find this out by relating the dominant frequency of H. eximia (98Hz) of Table 1 to this figure but unsuccessfully so (values here are around 3KHz).
Figures 2,3: Looking at these I find it hard to believe that the DFA scored 57% of these calls correctly according to basin of origin.
Figure 4. “Significant differences are indicated by different letters above the bars”. Differences to what? These are mean-with-error plots, I don’t understand which test of difference these correspond to. And what exactly do the letters a, b, c, and ab tell the reader about significance?
Figure 5: The correct abbreviation for cytochrome b gene is cob not cytB, the correct abbreviation for the rhodopsin gene is Rhod. None of the clades are statistically supported, so a bifurcating phylogeny is not an appropriate way to depict relationships among the individuals (also see Figure 6). A specimen table with locality information is missing. Genbank numbers need to be obtained (instead of providing an alignment in text format as In Appendix A).
Figure 7 & 8. “Maximum Likelihood (y-axis) was reached when the average K=3 groups” ?????

Figure “Phylogeographic Hypotheses”: The allopatric populations are indicated by an image of the frog that on the map looks either way, whereas in the legend it faces the right, and the sympatric populations face left. The figure would be better understandable if the allopatric frog symbols would all face to the right side.

Reviewer 4 ·

Basic reporting

No Comments

Experimental design

No comments

Validity of the findings

The revised version does a much better job to communicate the findings of the study. Despite that I somewhat disagree with one of the methods used (partial Mantel test), the authors cited some literature supporting their choice of analysis. I accept that the use of certain statistical methods is a "matter of taste", so I think my initial criticism has been sufficiently addressed.

Additional comments

The revised version does a much better job to communicate the study's findings. The command of English and formal errors have been much improved upon. Also the wording of the title and reduction of speculations greatly helped to improve clarity of the article. I have only some minor comments:

Abstract - coexistence with sister (.......) or with closely related species --> coexistence with sister species (...) or with

29ff: morphologic, ecologic, physiologic --> morphological, ecological, physiological

37: often good models --> remove "often"

58: Hyla have often been taken as models --> include references

156: remove random sub-header "Phylogeny; DNA..."

221: remove paragraph between second and third sentence of this section

Table 3 title: H. arenicolorand --> H. arenicolor and

Figure 3. I believe the final version of the paper will only be online and in color; but please think about using alternative symbols in this plot - the H. e. Panuco and H. e. Balsas point markers look identical when printed out black and white.

---

## Round 0.3 · accepted · Accept

Thanks for all your work on this ms which is now looking very nice.